

# Ultrasound and ultraviolet: crypsis in gliding mammals

Sasha L. Newar[1], Irena Schneiderová[2], Bryan Hughes[3] and Jeff Bowman[1,4]

[1] Environmental and Life Sciences, Trent University, Peterborough, ON, Canada
[2] Faculty of Science, Charles University, Prague, Czechia
[3] Faculty of Biology, Laurentian University, Sudbury, ON, Canada
[4] Ontario Ministry of Natural Resources and Foresty, Peterborough, ON, Canada

## ABSTRACT

Gliding is only present in six extant groups of mammals—interestingly, despite divergent evolutionary histories, all mammalian gliders are strictly nocturnal. Gliding mammals also seem to have relatively high rates of ultrasound use and ultraviolet-induced photoluminescence (UVP) in contrast with their close relatives. Therefore, we hypothesized that, despite diverging lineages, gliding mammals use similar modes of cryptic communication compared to their non-gliding counterparts. We developed two datasets containing the vocal range (minimum-maximum of the dominant harmonic; kHz) and UVP of 73 and 82 species, respectively; we report four novel vocal repertoires and 57 novel observations of the presence or absence of UVP. We complemented these datasets with information about body size, diel activity patterns, habitat openness, and sociality to explore possible covariates related to vocal production and UVP. We found that the maximum of the dominant harmonic was significant higher in gliding mammals when vocalizing than their non-gliding relatives. Additionally, we found that nocturnality was the only significant predictor of UVP, consistent with the previous hypothesis that luminophores primarily drive UVP in mammal fur. In contrast, however, we did not find UVP ubiquitous in nocturnal mammals, suggesting that some unknown process may contribute to variation in this trait.

## INTRODUCTION

Gliding mammals are physically unique: their most notable shared trait, a thin membrane (the patagium), stretches between limbs, digits, necks, and tails, allowing them to glide between trees and other elevated features in their arboreal habitats (*Jackson & Schouten, 2012*). Gliding has independently evolved at least nine times in mammals and is represented by six extant taxa (*Thorington & Heaney, 1981*; *Dudley et al., 2007*; *Jackson & Schouten, 2012*; *Jackson & Thorington, 2012*): colugos (Cynocephalidae; Dermoptera), flying squirrels (Pteromyini; Rodentia), scaly-tailed flying squirrels (Anomaluridae; Rodentia), lesser gliding possums (*Petaurus*; Diprotodontia), greater gliders (*Petauroides*; Diprotodontia) and the feather-tailed glider (*Acrobates*; Diprotodontia). Despite what some of their common names imply (*e.g.*, flying squirrels), gliding mammals are incapable of true flight, as exhibited by birds or bats. Instead, these mammals extend their patagium as they jump

Corresponding author
Sasha L. Newar,
sashanewar@trentu.ca

to convert gravitational velocity to forward momentum, allowing traversal of complex environments farther and faster than would occur through other means (*e.g.*, walking and climbing; *Dudley et al., 2007*; *Byrnes & Spence, 2011*). This locomotive advantage has been hypothesized to aid with predator avoidance (*Emmons & Gentry, 1983*), traversing vertical habitat structures (*Emmons & Gentry, 1983*; *Dial, 2003*), and improved foraging efficiency (*Paskins et al., 2007*; *Scheibe et al., 2007*).

While gliding mammals all share this unique locomotive trait, it is not the only trait they have evolved to share. All gliding mammals are strictly nocturnal (*Thorington & Heaney, 1981*; *Fokidis & Risch, 2008*; *Jackson & Schouten, 2012*); this is particularly notable in the flying squirrels, the only extant lineage of nocturnal squirrels (*Newar & Bowman, 2020*). Interestingly, unlike other gliders, which share some close nocturnal relatives, phylogenetic reconstructions suggest that traits associated with diurnality are the ancestral state of all squirrels (*Menéndez et al., 2021*), meaning that the emergence of nocturnality and gliding are entangled in flying squirrels. Even the oldest flying squirrel fossil, which was estimated to have originated 11.6 mya, already had well-developed wrist spurs (a key adaptation that both keeps the patagium tucked when climbing and holds the patagium open to increase surface area when gliding) and large orbital processes like current nocturnal species (*Casanovas-Vilar et al., 2018*). For the only other group of volant mammals, bats, it is also unclear if nocturnality evolved first, as previously assumed, or if nocturnality emerged in tandem with flight (*Anderson & Ruxton, 2020*). So, while the order in which these traits evolved in gliding mammals may be unclear, nocturnality appears to be an important covariate of volancy in mammals.

In bats, echolocation has evolved to aid in navigating complex environments and capturing moving prey in mid-flight. Echolocation is the (often) rapid production of calls that echo off solid objects, allowing the caller to interact with their environment in poor light conditions (*Panyutina et al., 2017*). The frequency of calls used to echolocate can vary from auditory (within the human auditory range from approximately 20 Hz to 20 kHz; *Masterton, Heffner & Ravizza, 1969*) to ultrasonic (>20 kHz). Most echolocating mammals commonly use ultrasonic vocalizations (USVs), including cetaceans, most bats, and small terrestrial mammals with poor eyesight, because the shorter wavelengths of USVs allow for greater detection accuracy (*Panyutina et al., 2017*). The evolutionary relationship between echolocation and volancy in bats is complex, with at least six competing hypotheses relating to the evolutionary past of these traits (*Anderson & Ruxton, 2020*). The most widely accepted of these hypotheses is that bats began as gliders, with echolocation likely developing in tandem as gliding evolved to the more complex behavior of flying (*Anderson & Ruxton, 2020*). Interestingly, several gliding mammals have also been shown to produce USVs, including colugos (*Miard et al., 2019*), feather-tailed gliders (*Martin, 2019*), and flying squirrels (*Gilley, 2013*; *Murrant et al., 2013*; *Gilley et al., 2019*; *Diggins et al., 2020*). While the function of these calls is still unclear, due largely to relatively few *in-situ* studies of mammalian gliders, there has been some evidence that gliders produce high-frequency (>10 kHz) calls in tandem with gliding behaviors (*Miard et al., 2019*). Additionally, flying squirrels produce significantly higher calls than non-gliding squirrels, suggesting that gliding may be linked to USV production (*Newar & Bowman, 2020*). However, nocturnality

and gliding are entangled in Sciuridae, with all nocturnal squirrels belonging to the flying squirrel tribe, Pteromyini, meaning that it is unclear whether nocturnality or gliding is more closely associated with high-frequency calls.

Aside from nocturnality and gliding, higher frequencies are also commonly associated with various other traits: smaller body sizes (*Martin, Tucker & Rogers, 2017*; *Newar & Bowman, 2020*) and open and structurally uncomplicated habitats (*Boncoraglio & Saino, 2007*; *Ey & Fischer, 2009*; *Fischer, Wadewitz & Hammerschmidt, 2017*). Additionally, some primates have demonstrated increased sensitivity to higher frequencies as social complexity increases (*Ramsier et al., 2012*) and a wide range of small mammals can exploit high frequencies for social contexts (*Arch & Narins, 2008*), such as alarm calling that is undetectable by focal predators (*Wilson & Hare, 2006*). These higher frequency calls are not restricted to the ultrasonic range, with birds, anurans, and mammals all displaying frequency shifts within the sonic range. While gliding mammals are relatively small-bodied and exhibit social behaviors, they exclusively inhabit forested habitats and even the largest gliders still need to navigate closed canopies. However, forested habitats should greatly restrain USVs as these environments easily attenuate high-frequency sounds. Yet, bird songs are acoustically complex despite forested habitats (*Boncoraglio & Saino, 2007*), and squirrels in open habitats only have a slightly higher peak frequency than those in closed habitats, with no effect on the maximum frequency of the dominant harmonic (*Newar & Bowman, 2020*). Additionally, bats foraging along edge habitats and within narrow spaces use higher frequencies than bats foraging in open habitats to optimize echolocating behavior (*Schnitzler, Moss & Denzinger, 2003*).

Flying squirrels have been shown to exhibit another unique trait: ultraviolet-induced photoluminescence (UVP) in fur (*Kohler et al., 2019*; *Reinhold et al., 2023*; *Toussaint et al., 2023*). Recent interest in this topic has led to the discovery of UVP in several species, including other gliding mammals (*Reinhold, 2021*) and relatives, such as springhares (*Olson et al., 2021*; shared suborder with scaly-tailed flying squirrels) and dormice (*Nummert, Ritson & Nemvalts, 2023*; shared suborder with flying squirrels). UVP occurs when ultraviolet (UV) light from the environment is absorbed and then re-emitted as visible light by excited particles, which, in the case of mammals, can be expressed in the fur (*Kohler et al., 2019*; *Reinhold, 2021*), quills (*Hamchand et al., 2021*), scales (*Jeng, 2019*), and teeth and bone (*Levin & Flyger, 1973*). It has been proposed that porphyrins and tryptophan metabolites (henceforth luminophores), both of which are known to photoluminesce under UV light and are ubiquitous across mammals, are the likely cause of UVP in the fur of some mammals (*Nicholls & Rienits, 1971*; *Olson et al., 2021*; *Hughes et al., 2022*; *Toussaint et al., 2023*). There is some evidence to suggest that most photoluminescent fur is the result of photodegradable porphyrins (*Toussaint et al., 2023*). However, tryptophan metabolites that are often associated with vivid fluorescent pigments in possums and some diurnal animals are not as readily photodegradable as porphyrins (*Pine et al., 1985*; *Schäfer, Goddinger & Höcker, 1997*; *Toussaint et al., 2023*). Thus, we might expect a greater prevalence of UVP in nocturnal species compared to diurnal species that experience increased photodegradation of porphyrins and lack sufficient concentrations of tryptophan metabolites within their fur. However, melanin can mask the photoluminescent properties of luminophores found
in mammal fur (*Huang et al., 2006*); therefore, mammals with darker fur should exhibit weaker or no UVP compared to mammals with lighter fur (*Rebell, 1966*). Notably, while the fundamental processes associated with UVP are understood in some species, there has yet to be a comprehensive review of which mammals exhibit (and perhaps more importantly, do not exhibit) UVP.

While empirical evidence demonstrating the behavioral relevance of UVP in mammals has yet to be presented in the literature, this trait has received substantial media attention, with several hypotheses aiming to describe a behavioral function. *Marshall & Johnsen (2017)* suggested the following criteria to conceptualize the communication potential of photoluminescent coloration in any taxa: visible location of colors, wavelengths of excitement and emission, viewer sensitivity, behavioral changes regarding photolumination, and natural light availability. While some mammals may exhibit UVP internally (fox squirrels exhibiting UVP in their bones (*Levin & Flyger, 1973*)), UVP in fur is easily visible to potential viewers. There is a broad excitation range for visible-spectrum photoluminescence emission, with excitation spectra from ~320–650 nm (*Huang et al., 2010*; *Hamchand et al., 2021*). While many nocturnal mammals are sensitive to the ultraviolet portion of this range *via* short-wave cone sensitivity ~360 nm (*Gerkema et al., 2013*), photoluminescence emission can occur as almost any color in the visible spectrum. Therefore, UV sensitivity is not necessarily required for UVP to be biologically relevant and instead, UVP is restricted by the availability of the environmental UV light to excite the photoluminescent structures. UV light drastically decreases during the night which suggests that nocturnal mammals have a lower potential for UVP to be relevant compared to diurnal mammals. However, UV reflectance of moonlight has been shown to change the relevance of UVP in some nocturnal non-mammalian species (*Kloock, 2005*; *Marshall & Johnsen, 2017*). Nocturnal species also lack the UV-filtering lens present in diurnal mammals, potentially allowing for a larger color range of UVP to be seen when there is enough UV light to cause UVP (*Yolton et al., 1974*).

Given the strong relationship between high-frequency sound production and gliding in squirrels and the recent discovery of UVP in flying squirrels, we wanted to further investigate these traits across all gliding mammals. The link between nocturnality and gliding in mammals allowed these species to exploit a particular niche; the communication methods used by nocturnal gliders might be constrained by the features associated with this niche. For example, gliding mammals are exposed to fewer predators than their diurnal relatives, but their predators are specialized for nocturnal prey detection (*Jackson & Schouten, 2012*). Owls are common predators of North American flying squirrels (*Glaucomys*) and employ large, low-light sensitive eyes to aid in prey detection (*Dice, 1945*). At the same time, owl ears are adapted for detecting low-frequencies (*Knudsen, 1981*), which would be advantageous for detecting movement-related sounds. Therefore, it would be beneficial for flying squirrels, which are socially complex species, to communicate with conspecifics at a frequency higher than what an owl is specialized to receive. Additionally, given the communication potential of UVP, we might expect UVP to be used either as a visual cue to conspecifics or as a conspicuous visual camouflage (*e.g.*, Batesian mimicry; *Kohler et al., 2019*), as owls are another known group to exhibit UVP (*Weidensaul et al., 2011*).

Crypsis is the ability of an animal to avoid detection by other animals, including visual (*Vignieri, Larson & Hoekstra, 2010*; *Ruxton et al., 2018*) and auditory (*Ruxton, 2009*; *Legett, Hemingway & Bernal, 2020*) concealment, such as using camouflaged signals or signals outside of the perceptual range of a predator (*Marples, Kelly & Thomas, 2005*). We consider the use of high-frequency communication and UVP as potentially cryptic traits because of evidence or hypotheses that the traits might be camouflaged or outside of the perceptual range of predators. Given the potential vulnerability of gliding mammals to predators and the apparent selective pressure toward nocturnality of the gliding trait, we were interested in the potential that crypsis was widespread among this group of species. Given their unique ecological niches and evolutionary pressures, we hypothesized that gliding mammals are more likely to exhibit these potentially cryptic traits than their close phylogenetic relatives. We selected a range of squirrels (to contrast with flying squirrels), rodents (to contrast with the scaly-tailed flying squirrels), primates (to contrast with colugos), and marsupials (to contrast with marsupial gliders) with similar body sizes to compare UVP and vocal ranges across gliding and non-gliding mammals. We predicted that physical (body size), behavioral (sociality, nocturnality), and environmental (habitat openness) traits would impact vocal range across all species, but higher frequencies would be most associated with gliding mammals. In contrast, given the current limited understanding of UVP, we did not expect UVP to be strongly associated with physical or environmental variables. UVP may also play a role in the communication of social species inhabiting visibly difficult or low-light environments. However, given the photodegradability of some luminophores that accumulate in fur and the communication potential of UVP in some nocturnal species, we also predicted that the pink-orange-red UVP would be strongly associated with nocturnality and sociality. Additionally, while we expected to find UVP in all gliding mammals, we predicted that UVP would be found in most nocturnal mammals tested. Our overall aim was to investigate the relationship between acoustic (vocal range) and visual (UVP) traits in gliding mammals in contrast with related species.

## MATERIALS & METHODS

### Vocalizations

We developed a database beginning with a list of publications describing gliding mammal vocalizations (summarized in Table S1). The minimum requirement for each publication was describing at least one call with either a spectrographic analysis or numerical data. However, most publications described multiple call types per species or multiple species per publication (seven gliding mammals represented across nine publications, summarized in Table S1). The databases used to search for these publications were Google Scholar, JSTOR, Web of Science, and Wiley Online Library. We used the keywords acoustics, acoustic repertoire, calls, frequency, Hz, vocalizations, and ultrasound paired with an exhaustive list of currently valid and invalid genera (the most updated nomenclature was taken from the Integrated Taxonomic Information System http://www.itis.gov/). Across all published calls, we took the absolute minimum and maximum frequencies (kHz) of the dominant harmonic for the final analyses (this often corresponded to the fundamental harmonic, if

multiple harmonics were present; following *Newar & Bowman, 2020*). For noisy calls, such as broadband calls, where the harmonics are not well defined, we estimated the minimum and maximum of the loudest parts of the call. We did not include calls produced by neonates or juveniles as there is evidence of some frequencies and calls being different in younger individuals (*Nikol'skii, 2007*; *Schneiderová et al., 2015*).

To compare gliding mammals to closely related species, we systematically searched for vocalization data using the same methodology described above (Fig. 1). Flying squirrels are unique amongst the gliders as there are many extant species that occupy the same family (Sciuridae); therefore, we kept all relatives from the same subfamily (Sciurinae) and a random subset of squirrels from the other subfamilies (26 squirrels across 62 publications). Other gliders have few extant relatives and we strategically chose taxa that shared similar evolutionary histories and traits. For the scaly-tailed gliders, we selected springhares (*Pedetes capensisi*), the only other extant taxa of the Anomaluromorpha suborder, and a variety of small-bodied rodents (12 species across 16 publications) exhibiting a range of vocal frequencies (maximum dominant frequency range: 9.86 (*Sicista subtilis*; *Volodin et al., 2019a*) - 109.8 kHz (*Mus musculus*; *Hoffmann, Musolf & Penn, 2012*)). For colugos, the only extant members of the order Dermoptera, we selected tree shrews (*Tupaia belangeri*) which form a sister clade with Dermoptera (*Nie et al., 2008*) and similarly sized taxa from the order Primates (19 primates across 27 publications), which are the next closest sister taxa (*Beard, 1993*). For marsupial gliders, we expanded our search to include similarly sized taxa of the order Diprotodontia as there were few records of marsupial vocalizations (5 marsupials across 6 publications). The vocalization data for two marsupial gliders (*Petaurus breviceps* and *P. norfolcensis*) and two glider relatives (*Pedetes capensis* and *Pseudocheirus peregrinus*; Fig. 1) were not available in the literature, and we worked with co-authors and collaborators to develop novel call descriptions for our study (methods in Article S1). We also provide vocalization data from free-ranging *Petaurus australis* (methods in Article S1) to opportunistically contrast our recordings with previously reported calls in the literature (*Kavanagh & Rohan-Jones, 1982*; *Whisson et al., 2021*). In the literature, four species were represented by a single subspecies only: *Otolemur garnettii lasiotis*, *Petaurista alborufus lena*, *Sciurus aberti kaibabensis*, and *Sciurus niger rufiventer*.

## Ultraviolet-induced photoluminescence

To expand on our vocalization dataset, we assessed the UVP of pelage for 83 species. Previous literature accounted for 19 species in our dataset; we sampled an additional 64 species from the mammal collections at the Canadian Museum of Nature and the Royal Ontario Museum (one mounted specimen (*Sicista subtilis*), otherwise all dry-preserved pelts; specimen and museum information provided on Dryad). We sampled species from the vocalization dataset preferentially. However, we opportunistically added ten species (bold type in Table S1) to increase the sample size of luminescing species. We followed the same vocalization protocol detailed above for both opportunistic and previously published UVP species; we found vocalization data for eight additional species (four opportunistic and four from previous UVP literature).

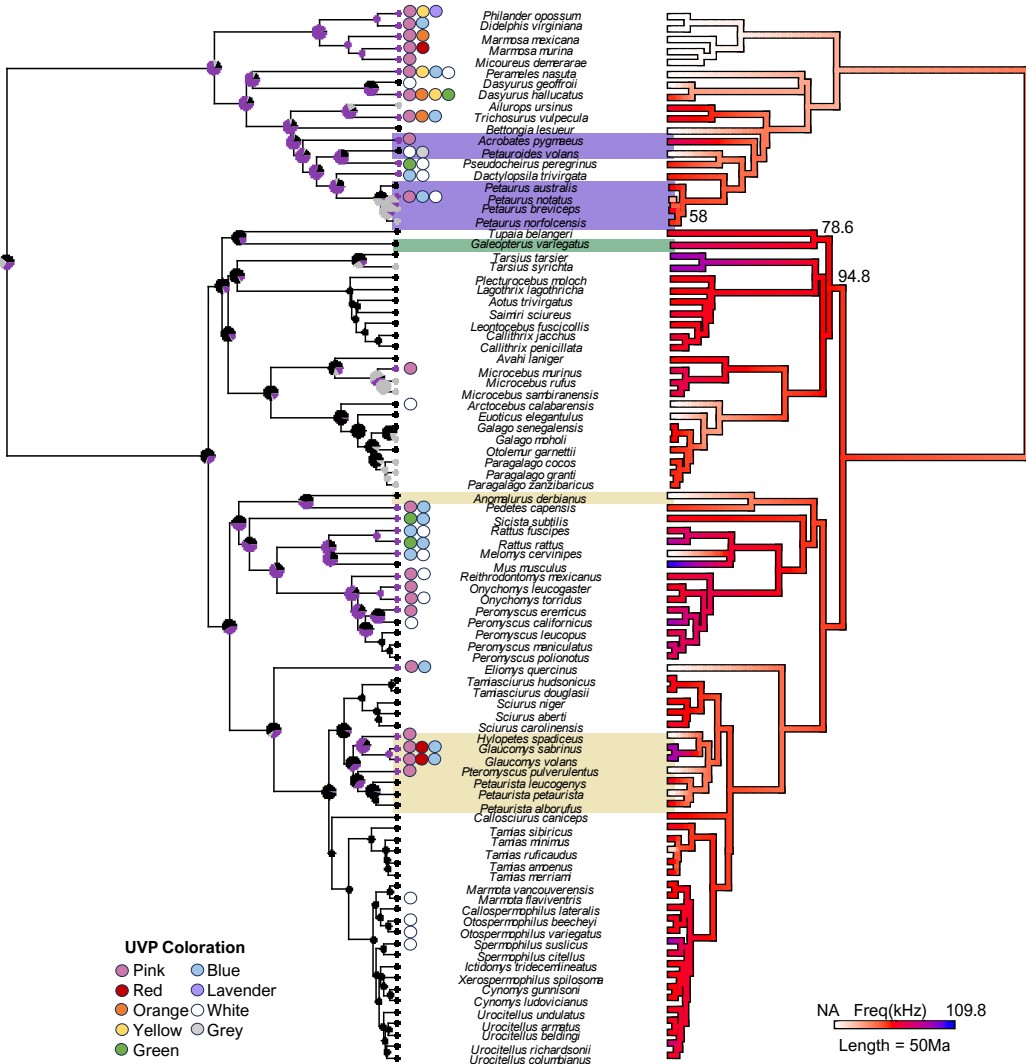

**Figure 1** **The final consensus tree representing traits associated with ultraviolet-induced photoluminescence (UVP) and vocal range limits (kHz) in gliding mammals and their relatives (*n* = 92).** Phylogeny estimated from mean edge lengths across 1,000 trees; bootstrap values <100 represented on the right phylogeny. The left stochastic character map represents UVP presence (purple = yes, black = no, grey = untested) with marginal frequencies at the nodes; circles along the left tips represent dominant UVP coloration. The right maximum likelihood map represents the character history of the maximum frequency (kHz; [0.12-84]) with white indicating species without vocal data (NA = 19). Gliding mammals are highlighted (marsupials = purple, colugos = green, rodents = yellow; *n* = 15; *Petauroides* included); scale bar represents 50 Ma.

We used a Vansky UV flashlight (395 nm wavelength) to illuminate museum specimens (held 75 cm above the individual) and a Huawei P30 Pro phone (held directly beside the light) to capture any luminescence. A yellow gel filter was held in front of the camera lens to reduce the input of purple-blue light (*Kohler et al., 2019*; *Nummert, Ritson & Nemvalts, 2023*). To minimize the additional yellow hue created by the filter, we manually color-corrected the photos in Photoshop (details in Info. S1). We took pictures of each
specimen's ventral and dorsal sides under white-light conditions, UV-light only, UV-light + filter, and UV-light + filter + correction (example provided in Fig. 2; complete photoset available on Dryad). We additionally photographed a few live *Glaucomys* individuals trapped in the Kawartha Highlands, Ontario, following the same protocol (animals studied under Trent University animal care protocol 27909).

In our investigation, some species expressed visible photoluminescence in white pelage or in some cases, the white ends of guard hairs. While "white" UVP has been noted in some species, this "white" coloration has been reported as a bluish-white (as seen in the striped possum (*Dactylopsila trivirgata*) and some marsupial gliders; *Reinhold, 2021*). The underlying cause of UVP expressed as distinct colors have been linked to porphyrins (red or pink) or tryptophan metabolites (cyan, blue, lavender; *Reinhold et al., 2023*). However, the expression of exclusively "white" coloration is not commonly reported, nor has a clear explanation been proposed for producing UVP without a dominant color. Furthermore, white human hair may emit a bluish hue similar to the pelage of minks, rabbits and goats and sheep, which have been described as being photoluminescent due to the presence of tryptophan metabolites (*Millington, 2020*). Given that we could not photograph museum specimens in complete darkness, the available visible light may have excited white hairs that would otherwise not express UVP. Therefore, to remove the potential bias of visible light, we removed individuals that only expressed "white" photoluminescence (but model outputs for all species, including those with "white" UVP, are included in Table S2). While UVP varied dramatically in color (*e.g.*, pink, blue/green), placement, and patterning across museum specimens and published literature, we reduced variability to absence/presence to increase the sample size in each category.

## Dataset assembly

Once we had assembled our database of vocalizing mammals with UVP records, we searched for the body mass (g), diel activity pattern (diurnal or nocturnal), social complexity, and habitat openness of the dominant habitat (open or closed) of each species. We preferentially took these data from the relevant vocalization or UVP papers, though this information was rarely provided; therefore, other resources, including articles and online databases such as Mammalian Species accounts and the Animal Diversity Web (*Myers et al., 2023*), were reviewed to complete our dataset. If a range was provided for the body mass, we took the mean of the given values; we took a mean of male and female body masses as we were not capturing the effect of sex on vocalization frequencies or UVP. Social variability was reduced to social or solitary living to reduce model parameters; species that exhibit dynamic social structures, where adult individuals will seasonally or cyclically shift between solitary and social living (*e.g.*, flying squirrels engaging in social nesting during the winter only; *Garroway, Bowman & Wilson, 2013*), were treated as socially living.

## Phylogeny

While multiple subspecies were present in the vocalization dataset, we calculated the vocalization maxima at the species level for the final dataset and analyses (Fig. 1; subspecies-specific information noted in Table S1). Only one subspecies was excluded from analyses

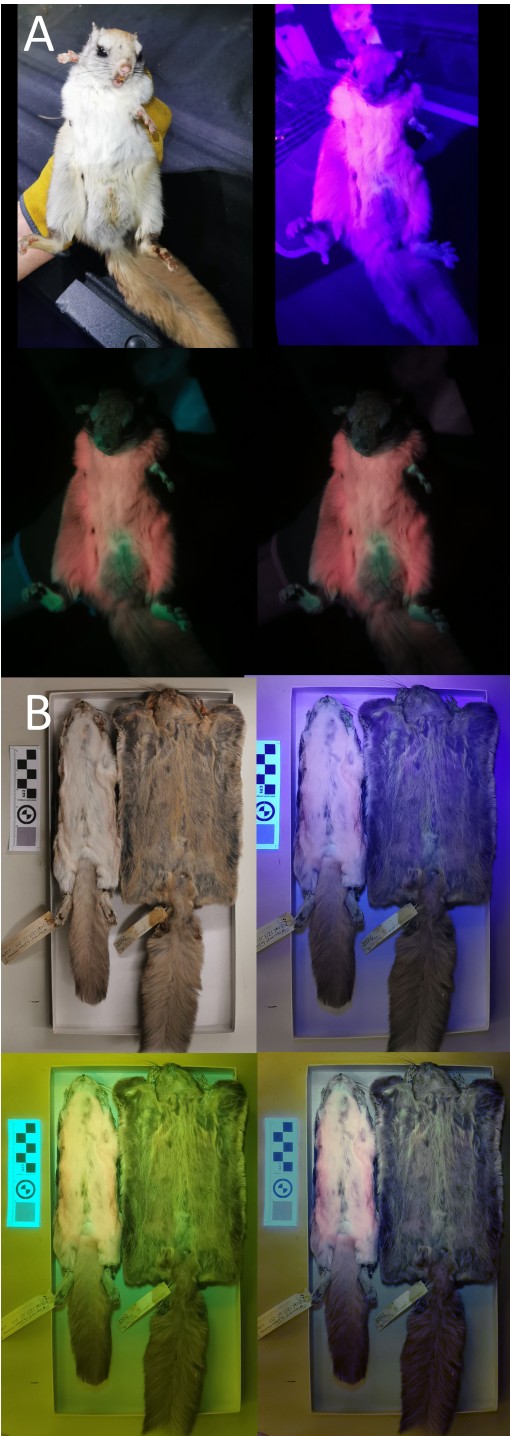

**Figure 2** **Variation in ultraviolet-induced photoluminescence demonstrating the variability within museum specimens and between live and preserved individuals.** (A) Live adult male trapped in the Kawartha Highlands in 2023. (B) Dry-preserved museum specimens from the Canadian Museum of Nature. Top left: white light; top right: ultraviolet light; bottom left: ultraviolet light and yellow gel filter; bottom right: ultraviolet light, yellow gel filter, and color edit (Supplementary Information S1).

(*Peromyscus maniculatus bairdii*) due to a binary variable inconsistency with the parent species—this subspecies only occurs in open habitats (*Wecker, 1963*), while the parent species is most commonly found in closed habitats. In addition, *Masters et al. (2017)* recently proposed that the *Paragalago* genus is a distinct clade from the *Galagoides* genus to which the *Paragalago* species had been previously assigned; we reassigned these species accordingly.

For the final species dataset ($n = 93$), we pruned 1,000 node-dated completed trees from the mammalian supertree on VertLife, an online database used to produce pruned random distribution trees of vertebrate species (*Upham, Esselstyn & Jetz, 2019*). The nexus outputs were compiled into a consensus tree using the *phytools* (*Revell, 2012*) package in R (*R Core Team, 2022*) (Fig. 1). *Petaurus notatus* is a recently described species (previously incorporated within *P. breviceps*), and therefore, it was the only species not available on Vertlife; we incorporated this species into the final consensus tree by splitting the *P. breviceps* lineage at a divergence time of 1 Ma (*Cremona et al., 2021*).

## Analyses

We built phylogenetic generalized least square (PGLS) models to account for variation in the vocal range that could be explained by phylogenetic relatedness. PGLS models estimate phylogenetic relatedness as lambda ($\lambda$), which varies between 0 (no phylogenetic trace) and 1 (absolute Brownian motion) (*Freckleton, Harvey & Pagel, 2002*; *Martin, Tucker & Rogers, 2017*). Full models were built for each frequency limit ($\beta_0$ + body mass ($\beta_{Mass}$) + diel activity pattern ($\beta_{Diel}$) + sociality ($\beta_{Sociality}$) + habitat openness ($\beta_{Open}$) + UVP ($\beta_{UVP}$)) using the *caper* (*Orme et al., 2018*) package in R (*R Core Team, 2022*). We reported the regression coefficient estimates ($\bar{x} \pm SE$) to evaluate significance and effect size (*F*-statistic, *P*-value, and adjusted $R^2$).

We also built a phylogenetic generalized linear mixed (PGLM) model for binary data using the *ape* (*Paradis & Schliep, 2019*) package in R to assess the presence of UVP. This binary PGLM model accounted for variation in UVP while dealing with the bimodal distribution that violates other tests (*Ives & Garland, 2010*). The same independent variables were used ($\beta_0$ + $\beta_{Mass}$ + $\beta_{Diel}$ + $\beta_{Sociality}$ + $\beta_{Open}$); $\beta_{Mass}$ was standardized to have a mean of 0 and variance of 1, while the categorical variables were reconstructed into dummy variables (2 categories = 0, 1) for the PGLM model. We standardized the variables to improve the interpretation of regression coefficients as they more accurately represent the effect size of the independent variables (*Ives & Garland, 2010*). The PGLM model represents the phylogenetic signal ($s^2$) as the scalar magnitude of the phylogenetic variance across all species comparisons (estimated from the phylogenetic variance–covariance matrix; (*Ives & Garland, 2010*).

## RESULTS

### Phylogeny

Our final phylogeny (Fig. 1) contained 92 species from three mammalian lineages: primates and relatives (Dermoptera, Scandentia, and Primates), rodents (Rodentia), and marsupials (Diprotodontia). Stochastic character mapping with marginal frequencies of UVP and

maximum likelihood of the maximum frequency (kHz) projected onto our phylogeny (estimated from 1,000 simulations each) supported the hypothesis that high-frequency communication is species-specific (with high-frequencies only showing up on branch tips; Fig. 1). This was further supported by the weak phylogenetic signal (λ [95% CI]) detected for the minimum frequency (0 [0, 0.40]) and variable phylogenetic signal for the maximum frequency (0.77 [0, 0.95]). Interestingly, we found a significantly stronger phylogenetic signal for UVP ($s^2 = 0.39$, $p <0.001$), suggesting that UVP is not species-specific and is, instead, a broader taxonomic trait. This is demonstrated in the phylogeny, where primates rarely exhibit UVP, most rodents (except for diurnal squirrels and some *Peromyscus*) exhibit UVP, and UVP is variable among marsupials. Despite this variation among the orders, UVP appears in the marginal frequencies of all nodes for the first ~100 Ma, further supporting the finding that UVP is likely ancestral to some extent in most mammals.

## Vocalizations

Our final vocalization dataset consisted of 73 species, of which nine were gliding mammals. In this publication, we contributed call descriptions for five species (*Pedetes capensis, Petaurus australis, Petaurus breviceps, Petaurus norfolcensis,* and *Pseudocheirus peregrinus*), four of which have not been published previously (*Petaurus australis* has been previously reported; Table S1). We found that our collaborator recorded calls consistent with previous literature for *Petaurus australis*, which provides confidence for our novel descriptions reported here. All five species were recorded with microphones sensitive to the human auditory range (20 Hz–20 kHz); however, we also opportunistically recorded sugar gliders (*Petaurus breviceps*) with ultrasonic detectors. We found that sugar gliders produced at least one truly ultrasonic call type along with two calls that extended into the ultrasonic range (bark, broadband burst) and three additional high-frequency calls (>10 kHz) that could be detected on the ultrasonic microphones (high frequency, sniffing, whistle; Fig. S1 & Table S3). The ultrasonic microphones used can distort calls in the sonic range, therefore, the values presented in Table S3 should be further investigated with sonic microphones. Additionally, given that we were unable to remove pups from the recording space, we did not include the call type 'ultrasonic' in the final analysis as they were rare and had very low amplitude; additionally, many mammalian pups produce ultrasonic isolation calls that are lost (or decrease in frequency) later in life. Therefore, we conservatively removed the ultrasonic call type from our dataset. In addition to the high-frequency calls, sugar gliders also produced a low-frequency vocalization (yap) similar to the other marsupials recorded (Fig. S2 & Table S4) Of the four additional species recorded, non-gliding springhares (*Pedetes capensis*) had the most consistent repertoire, with only one confirmed call type (growl) recorded over 100 times (Fig. S2 & Table S4). Yellow-bellied gliders (*Petaurus australis*) produced the longest vocalizations, with almost all calls lasting longer than one second (Fig. S2 & Table S4).

Across all species in the dataset, we found that body size had a negative effect on the minimum (Estimate: $\bar{x}$ ($\pm SE$) $= -0.48 \pm 0.15$; Effect Size: $F_{df} = 22.07_{1,56}$, $p <0.001$) and maximum ($-0.39$ ($\pm 0.09$); $F_{df} = 19.48_{1,56}$, $p <0.001$) frequencies. Additionally, gliding was positively related to the maximum frequency (1.14 ($\pm 0.50$); $F_{df} = 6.81_{1,56}$, $p$

= 0.01). We also found that nocturnal mammals produced significantly higher minimum frequencies than their diurnal counterparts (1.32 ($\pm$0.62); $F_{df} = 5.10_{1,56}$, $p = 0.03$), as did solitary mammals (1.01 ($\pm$0.48); $F_{df} = 4.20_{1,56}$, $p = 0.05$). Additionally, we did not detect a relationship between UVP and either frequency limit.

### Ultraviolet-induced photoluminescence

Our final UVP dataset consisted of 82 species, of which 27 were photoluminescent under UV conditions (16 from literature, 11 novel reports). We found an additional seven species which expressed strictly white UVP which were treated as non-photoluminescent in the analysis (Fig. 1). Nocturnality was the only trait with a significant relationship to UVP, with nocturnal species significantly more likely to exhibit UVP (Estimate: 8.09 ($\pm$3.79), $p = 0.03$). However, despite nocturnality being a significant covariate, we found that a similar number of nocturnal species did not exhibit UVP, with 23 species not exhibiting UVP compared to the 26 that did (blue–green-yellow-pink). Out of the nocturnal mammals that expressed UVP, 20 displayed pink (occasionally with orange or red) photoluminescence while 10 expressed blue or green, with five of these species expressing both blue and pink (Fig. 1). We did not find any instances of UVP in the diurnal mammals used in this study (Fig. 1). We also found that gliding, sociality, size, and habitat openness were not associated with the absence or presence of UVP. Finally, unlike the vocalization data, we found a significant phylogenetic signal for UVP ($s^2 = 0.39$, $p < 0.001$).

## DISCUSSION

In support of our hypothesis, we found that gliding mammals exhibited significantly higher vocal ranges than their non-gliding counterparts. We were also able to demonstrate high-frequency calls in sugar gliders for the first time, which further supports our finding that high-frequency communication is common across gliding mammals. Despite discovering UVP in several new species, we found no relationship between UVP and vocal limits or gliding, despite some flying squirrels and glider relatives exhibiting both traits. However, we found a significant relationship between nocturnality and non-white UVP, further providing evidence for the hypothesis that nocturnal species will exhibit UVP because of the accumulation of porphyrin luminophores (*i.e.*, pink-orange-red photoluminescence) without photodegradation observed in diurnal species (*Toussaint et al., 2023*).

### Vocalizations

As predicted, we found that the capacity to emit high-frequency vocalizations is a common trait across gliding mammals. We recorded high-frequency vocalizations in sugar gliders (Fig. S1), which is the first record of calls reaching ultrasonic frequencies in marsupial gliders. Notably, despite being unable to test for USVs in other marsupial gliders (*i.e.,* yellow-bellied or squirrel gliders), we still found that gliding was one of only two key traits significantly associated with higher maximum frequency use. Body size is a common predictor for vocal limits and has been explored across various taxa (*Ryan & Brenowitz, 1985*; *Evans, Neave & Wakelin, 2006*; *Pfefferle & Fischer, 2006*; *Cui, 2012*; *Charlton & Reby, 2016*; *García-Navas & Blumstein, 2016*; *Martin, Tucker & Rogers, 2017*). Vocal limits are

**Table 1  Model results for the frequency range (minimum and maximum frequencies; kHz; $n = 73$) and ultraviolet-induced photoluminescence (UVP; $n = 82$) of gliding Mammalia and their relatives.** Phylogenetic least square (PGLS) models were conducted with the frequency ranges while a binary phylogenetic generalized linear mixed (PGLM) model was used for the UVP dataset. Effect size (Fdf, n; PGLS only) and slopes ($\bar{x} \pm$ SE) are estimated for each variable: gliding (Y/N), diel activity pattern (Diel A.P.; nocturnal/ diurnal), habitat openness (Habitat; Closed/Open), sociality (solitary/social), and UVP (Y/N). The phylogenetic signal (PGLS: λ [95% CI], PGLM: s2; estimated from 1,000 trees) and estimations of model fit ($R^2$, PGLS: F) are provided. Significant estimates are shown in bold.

| | Minimum frequency | | Maximum frequency | | UVP |
|---|---|---|---|---|---|
| | $\bar{x}$ ($\pm SE$) | $F_{df}$ | $\bar{x}$ ($\pm SE$) | $F_{df}$ | ($\bar{x} \pm SE$) |
| Intercept | 1.60 ($\pm$1.01), $p = 0.12$ | – | **4.56 ($\pm$0.99) $p <0.001$** | – | **−7.77 ($\pm$3.93), $p = 0.05$** |
| **log(Body mass (g))** | **−0.48 ($\pm$0.15) $p = 0.002$** | **22.07$_{1,56}$ $p <0.001$** | **−0.39 ($\pm$0.09) $p <0.001$** | **19.48$_{1,56}$ $p <0.001$** | −0.66 ($\pm$0.53) $p = 0.21$ |
| Gliding: $Y$ | 0.37 ($\pm$0.77) $p = 0.64$ | 3.75$_{1,56}$ $p = 0.06$ | **1.14 ($\pm$0.50) $p = 0.03$** | **6.81$_{1,56}$ $p = 0.01$** | −1.13 ($\pm$0.96) $p = 0.24$ |
| Diel A.P.: *nocturnal* | **1.32 ($\pm$0.62) $p = 0.04$** | **5.10$_{1,56}$ $p = 0.03$** | −0.13 ($\pm$0.48), $p = 0.79$ | 0.23$_{1,56}$ $p = 0.63$ | **8.09 ($\pm$3.79) $p = 0.03$** |
| Habitat: *open* | −0.23 ($\pm$0.48) $p = 0.63$ | 0.25$_{1,56}$ $p = 0.62$ | −0.29 ($\pm$0.29), $p = 0.33$ | 1.25$_{1,56}$ $p = 0.27$ | 0.39 ($\pm$0.99) $p = 0.70$ |
| Sociality: *solitary* | **1.01 ($\pm$0.48) $p = 0.04$** | **4.20$_{1,56}$ $p = 0.05$** | −0.46 ($\pm$0.24), $p = 0.06$ | 3.87$_{1,56}$ $p = 0.05$ | 0.72 ($\pm$0.75) $p = 0.34$ |
| UVP: $Y$ | −0.37 ($\pm$0.64) $p = 0.56$ | 0.34$_{1,56}$ $p = 0.56$ | −0.14 ($\pm$0.35), $p = 0.69$ | 0.16$_{1,56}$ $p = 0.69$ | |
| λ | 0.00 [0.00, 0.37] | | 0.82 [0, 0.96] | $s^2$ | **0.39, $p < 0.001$** |
| $R^2$ | 0.32 | | 0.29 | $R^2$ | 0.64 |
| F, P | **5.95$_{6,56}$, <0.001** | | **5.30$_{6,56}$, <0.001** | | |

highly controlled by vocal-producing structures that increase with body size and produce larger sound waves, perceived as lower frequencies (*Martin, Tucker & Rogers, 2017*). Therefore, we expected body size to be a significant predictor in our dataset. However, an important limitation of our study is that we did not separate ultrasonic and audible calls in our analysis. Ultrasonic calls are often produced *via* an aerodynamic whistle mechanism in the ventral pouch on the larynx (*Riede, Borgard & Pasch, 2017*; *Abhirami et al., 2023*), which may circumvent the negative body size relationship seen across most mammals. However, further investigation into the role and variation of the ventral pouch across many ultrasound-producing mammals would provide further insight into whether purely ultrasonic calls are also limited by body size.

The positive relationship between gliding and vocal limits was previously explored in the squirrel family (Sciuridae; *Newar & Bowman, 2020*), however, within Sciuridae, nocturnality and gliding are entangled traits, with gliding squirrels also being the only extant nocturnal species. Therefore, our current investigation, which incorporates gliding mammals and their nocturnal and diurnal relatives, allowed us to disentangle the nocturnal and gliding traits associated with all gliding mammals (*Jackson & Schouten, 2012*). When we modelled this larger dataset, we found nocturnality was positively related to increased minimum vocal frequencies while gliding was positively related to increased maximum vocal frequencies (Table 1).

The function of higher vocal limits in gliding mammals is likely complex, though these functions remain unclear, with few behavioral accounts linked to vocal recordings. USVs have been predominantly reported in echolocating mammals like bats and cetaceans (*Ahlén, 2004*; *Yovel et al., 2011*; *Parker et al., 2013*; *Thomas & Hahn, 2015*; *Carter & Adams, 2016*), with some hypothesizing that bats began as nocturnal gliders. Our investigation into cryptic communication in non-flying mammals supports this hypothesis, with gliding mammals producing USVs without a clear echolocation function. The first record of USVs in gliding mammals was in the North American flying squirrels (*Glaucomys*; *Murrant et al., 2013*), with subsequent reporting in the feather-tailed pygmy glider (*Martin, 2019*) and colugo (*Miard et al., 2019*) and novel USVs reported here in the sugar glider (Fig. S1 & Table S3). While the vocal repertoire of many previously and newly reported gliding mammals do not contain USVs (*Ando & Kuramochi, 2008*; *Shen, 2013*; *Poje, 2016*), these calls require specialized recording equipment, and the frequency limits of recording equipment are highly associated with the maximum frequency limits detected (*Newar & Bowman, 2020*). Given the recent reporting of USVs in multiple gliding mammals and the strong association between higher vocal limits and gliding reported here, we encourage researchers with access to these low-frequency gliders (including scaly-tailed flying squirrels, most marsupial gliders, and giant flying squirrels) to record individuals with ultrasonic equipment to determine if USVs are also present in these species. The presence (or absence) of USVs in other gliding mammals and any novel behavioral contexts for these calls should clarify the role of high frequencies in gliding mammals.

Given the currently available data, gliding mammals use significantly higher maximum frequencies than their non-gliding relatives. It may be intuitive to assume that these higher frequencies play an essential role in gliding, particularly given the predominant role of USVs in bats. Bats use USVs for echolocation, wherein USVs are rapidly produced to detect objects and often prey while both the individual and the object are moving (*Jones & Siemers, 2011*). This trait is highly specialized to both the vocal-producing structures and auditory receptors that must vibrate fast enough to produce and detect USVs (*Anderson & Ruxton, 2020*). The rate of call production is not nearly rapid enough in gliding mammals to mimic echolocation in bats (which varies between 2 and 20 pulses/s; *Jones & Siemers, 2011*). However, the frequency and production rate are like that of echonavigating shrews (*Gould, Negus & Novick, 1964*; *Tomasi, 1979*; *Siemers et al., 2009*) and blind mice (*Panyutina et al., 2017*; *Volodin et al., 2019b*), who use USVs to navigate complex spaces (*Siemers et al., 2009*; *Panyutina et al., 2017*). Both taxa have reduced vision, perhaps as a result of their dark environments, which may explain why acoustic signals have been selected as a navigation tool; like echolocation, echonavigating calls are produced to help orient an individual to their environment and do not seem to require the same physical specializations to be produced. Given the nocturnal behavior of all gliding mammals, which navigate complex, arboreal environments in reduced light conditions, similar selection pressures may have allowed for echonavigation to develop in this system. However, experiments like those shown in other echonavigating mammals (*Gustafson & Schnitzler, 1979*; *Siemers et al., 2009*) would be required to explore this hypothesis further.

Aside from echonavigation, many other mammals produce USVs for non-navigating purposes. Even within our dataset, 18 species produced calls at least partially in the ultrasonic range. The function of USVs in other species has been explored across several hypotheses, many of which were incorporated into our analyses. Interestingly, we found that habitat openness, which has been shown to be associated with higher frequency production, did not have a significant relationship with higher frequencies in our dataset. Previous studies exploring the role of habitat openness and attenuation of sound waves across the landscape have heavily biased their examples to open habitat species (*Koeppl, Hoffmann & Nadler, 1978*; *Blumstein, 2007*; *García-Navas & Blumstein, 2016*). Indeed, many species produce high frequencies and a variety of USVs in closed habitats despite these calls being easily absorbed by the spatially complex habitat in which they are produced. Furthermore, a truly subterranean rodent, the mole vole (*Ellobius talpinus*), has been shown to produce ultrasonic vocalizations (*Volodin et al., 2022*) despite the assumed acoustic restrictions of living underground, where acoustic signals are quickly absorbed by the dense surrounding environment. Therefore, while other researchers have predicted that open habitats may be more conducive to the evolution of USV production (*Boncoraglio & Saino, 2007*; *Ey & Fischer, 2009*; *Fischer, Wadewitz & Hammerschmidt, 2017*), it may be that open habitats are better for recording USVs (as previously proposed by *Newar & Bowman, 2020*). Sociality has also been previously investigated as a driver of vocal behaviors in mammals (*Hauser, 1993*; *Blumstein & Armitage, 1997*; *Arch & Narins, 2008*; *Ramsier et al., 2012*; *Faure et al., 2017*). In our dataset, we found that solitary mammals produced significantly higher minimum frequencies than their social counterparts; this is an interesting finding as literature points to social mammals using higher frequencies, with high-frequency hearing in primates increasing with vocal complexity (*Ramsier et al., 2012*) and ultrasound being used in a variety of social contexts in small mammals (*Arch & Narins, 2008*). However, *Hauser (1993)* demonstrated that frequencies can vary with different social encounters, particularly that frequencies decrease with aggressiveness and increase with fearfulness. We also found a nearly significant relationship between decreased maximum frequencies and solitary species. Therefore, our findings suggest that the solitary mammals in our dataset have less frequency variability than their social counterparts due to decreased social complexity.

## Ultraviolet-induced photoluminescence

We did not find that ultraviolet-induced photoluminescence (UVP) was associated with vocal limits, nor did we find that UVP was associated with gliding (Table 1). While UVP has been recently described in North American flying squirrels (*Glaucomys sabrinus* and *G. volans*), the Australian Krefft's glider (*Petaurus notatus*), the red-cheeked flying squirrel (*Hylopetes spadiceus*) and the smoky flying squirrel (*Pteromyscus pulverulentus*) we were unable to confirm UVP in any other gliding mammals (Fig. 1). However, we noticed substantial variation in the presence of UVP in North American flying squirrel museum specimens when confirming UVP presence in mammals known to express detectable levels of photoluminescence. We tested four dry-preserved museum specimens from *Glaucomys sabrinus* and *G. volans* each and observed very weak pink and blue UVP in one individual

from each species as well as considerable variation in the dorsal UVP across all eight individuals (the dorsal UVP is weaker in both species). Comparing UVP in dry-preserved *Glaucomys* specimens to live individuals, there is a striking difference in the strength and variation of colors observed under UV light, with live individuals producing very strong UVP coloration (Fig. 2). Therefore, not observing UVP in our study may be an artifact of poor preservation or specimen age rather than a lack of UVP. This key finding is important for researchers considering the use of museum specimens for UVP studies. Museum specimens are already more likely to produce false-positives due to the use of chemicals during preservation and mounting processes. This is the first study to directly compare live and dry-preserved individuals using the exact same methods and demonstrate that false-negatives are just as likely, if not more likely, to occur when using museum specimens. Similarly, due to the photosensitive nature of porphyrins, reddish photoluminescence is generally not expected to be retained in museum specimens (*Daher et al., 2020*). However, pink UVP was the most common color detected in our museum specimens, with 12 exhibiting pink compared to only two cases of blue and blue/green. Additionally, the pink UVP in the dry-preserved flying squirrels was much more pronounced than the blue, especially when contrasted against the live individuals (Fig. 2). We encourage researchers with access to other mammalian gliders (particularly giant flying squirrels, colugos, and marsupial gliders) to assess UVP with live specimens to either confirm a lack of UVP or to challenge our findings (*Reinhold et al., 2023*). We also note that the pelt preservation processes are unknown in the species used and there is a possibility that we have detected false positives due to chemicals and not natural photoluminescence, particularly in the mounted *Sicista subtilis.* Therefore, we further encourage other researchers to confirm investigate UVP in live-specimens whenever possible; developing a more comprehensive record of live-specimen UVP is crucial to understanding the ecological importance.

Despite our limitations, we found a significant relationship between nocturnality and the presence of UVP in our dataset. Several researchers have proposed that porphyrin induced UVP in fur should be highly associated with nocturnality (*Kohler et al., 2019*; *Olson et al., 2021*; *Toussaint et al., 2023*). Specifically, porphyrins and tryptophan metabolites, which readily accumulate in mammalian fur through various physiological pathways, have been identified as the main compounds associated with UVP in mammal pelage (*Toussaint et al., 2023*; *Reinhold et al., 2023*). Some of these luminophores are easily degraded by UV rays emitted by the sun (*e.g.*, porphyrins), while others are not as photodegradable (*e.g.*, tryptophans) and have been shown to cause UVP in some diurnal mammals. Additionally, heavy melanin loads in the fur can mask UVP in any mammal, regardless of temporality. Therefore, we had expected that there may be a greater prevalence of UVP in nocturnal species compared to diurnal species when there are low melanin loads in the fur. Consistent with this hypothesis, nocturnal species in our analysis with dark fur (*e.g.*, *Aotus trivirgatus*, *Otolemus garnettii*) did not exhibit UVP. However, we found considerable variation in UVP across nocturnal species concerning both occurrence (26 present, 23 absent including four white-only species) and coloration (from our methods: nine predominantly pink, one predominantly blue/green, three with mixed blue and pink coloration). This variation in UVP is greater than what we would expect if UVP is ubiquitous among nocturnal

mammals and suggests that the mechanism behind pelage UVP is likely more complicated than luminophore degradation.

Interestingly, UVP has been proposed as a method for social communication (*Kohler et al., 2019*). However, we found no evidence for UVP being associated with social species compared to solitary species. Looking at our data, half of the UVP species were solitary, including *Didelphis virginiana*, *Marmosa* spp., and *Peromyscus eremicus*. The role of UVP as a social trait is challenging to reconcile with our finding that several UVP species exhibit alternate social systems (*Garroway, Bowman & Wilson, 2013*), where they cyclically spend significant portions of their lives as solitary individuals. It remains possible that UVP can be a form of crypsis by contributing to visual camouflage (*Ruxton et al., 2018*; *Kohler et al., 2019*). Consistent with our hypothesis regarding predation avoidance in gliders, we found that six of the tested gliders exhibit pink UVP on their ventral pelage. Including the white-only UVP shown in *Petauroides volans*, seven of the 14 gliders exhibited UVP; notably, the gliders not exhibiting UVP in this study are all dry-preserved specimens while those gliders exhibiting UVP has been mostly confirmed in live or recently deceased individuals (with the exception of *Hylopetes spadiceus* and *Ptermyscus pulverulentus*). This further supports our finding that false negatives may be prolific in museum specimens and live animals should be used whenever possible.

## CONCLUSIONS

We found that gliding mammals emitted significantly higher vocal frequencies than their non-gliding relatives. Additionally, we found strong support for the role of body mass in reducing vocal frequencies across all taxa and no evidence for sociality, habitat openness, or UVP as key correlates of vocal limits. We propose that habitat openness and sociality may not be as crucial for predicting frequency limits as previously proposed. We contributed novel vocal repertoires for four species, and novel UVP reports for 57 species, of which 11 displayed non-white UVP. Finally, we found that nocturnality was the only significant predictor of UVP, with half of the nocturnal mammals tested exhibiting UVP of various colors (blue–green-yellow-pink). While UVP was not significantly more associated with gliding mammals compared to non-gliders, we also found that half of the gliders tested exhibited colored UVP. We conclude that gliding mammals have shifted to higher vocal frequencies to conceal themselves from potential eavesdroppers and while some gliders may be exploiting UVP to camouflage themselves in their environment, UVP is not as ubiquitous in gliders as high-frequency communication. While gliding mammals can be cryptic, their cryptic traits are likely just as influenced by their varied phylogenetic histories as they are by their convergent evolution.

## ACKNOWLEDGEMENTS

Thank you to Marc Anderson for access and permission to use novel marsupial vocalizations and J. McAdam for allowing us to record their sugar gliders. Thank you to Dr. Kamal Khidas at the Canadian Museum of Nature and Dr. Burton Lim at the Royal Ontario Museum for access to specimens in their mammal collections. Thank you to K. Solmundson,

B. Newar, T. Burgess, S. Kielar, and K. Martin for assistance with UVP photography; thank you to M. Bivi for your invaluable assistance with developing a standardized photo editing methodology. Finally, thank you to our three reviewers for their invaluable feedback during the editing and review process.

### Funding

This research was funded through an NSERC Discovery Grant to Jeff Bowman and Queen Elizabeth II Graduate Scholarships in Science and Technology awarded to Sasha L. Newar and Bryan Hughes. Additional support was provided by the Ontario Ministry of Natural Resources, Trent University, and Laurentian University. The funders had no role in study design, data collection and analysis, decision to publish, or preparation of the manuscript.

### Grant Disclosures

The following grant information was disclosed by the authors:
NSERC Discovery.
Queen Elizabeth II Graduate Scholarships in Science and Technology.
Ontario Ministry of Natural Resources, Trent University, and Laurentian University.

### Competing Interests

The authors declare there are no competing interests.

### Author Contributions

- Sasha L. Newar conceived and designed the experiments, performed the experiments, analyzed the data, prepared figures and/or tables, authored or reviewed drafts of the article, and approved the final draft.
- Irena Schneiderová conceived and designed the experiments, performed the experiments, authored or reviewed drafts of the article, and approved the final draft.
- Bryan Hughes conceived and designed the experiments, authored or reviewed drafts of the article, and approved the final draft.
- Jeff Bowman conceived and designed the experiments, authored or reviewed drafts of the article, and approved the final draft.

### Animal Ethics

The following information was supplied relating to ethical approvals (i.e., approving body and any reference numbers):

The Trent University Animal Care Committee provided full approval for this research.

### Data Availability

The data and code is available on Dryad: Newar, Sasha; Schneiderová, Irena; Hughes, Bryan; Bowman, Jeff (2024). Vocal range and ultraviolet-induced photoluminescence in gliding mammals and their relatives [Dataset]. Dryad. https://doi.org/10.5061/dryad.3n5tb2rp4.

## Supplemental Information

Supplemental information for this article can be found online at http://dx.doi.org/10.7717/peerj.17048#supplemental-information.

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
