# Peer review of "Ultrasound and ultraviolet: crypsis in gliding mammals"

_PeerJ, doi:10.7717/peerj.17048_

## Round 0.1 · original submission · Major Revisions

There was a wide split among the reviewers tor your paper. #1 focused on vocalisations and said Reject, #2 focused on UV and was mostly complimentary, but provided a large number of Minor revisions. #3 called for Major Revisions, and highlighted the state of the references.

Please place a special emphasis on revising/explaining the vocalisation data. Where reviewer #1 has pointed out issues with the sonic range of recording instruments or the included sonograms, do your best to meet these criticisms.

Both reviewers 1 and 3 ask that you clearly justify your choices of species. Please attend to this critique with special care.

Reviewer 1 ·

Basic reporting

The authors made a challenging work to provide a meta-analysis of literature and of some own data regarding the relationship of acoustic parameters and ultraviolet fur coloration with gliding/non gliding lifestyle and body size in a few dozen of mammalian species (including primates, rodents and marsupials).
I can only judge about the analysis of acoustic parameters of the vocalizatoons and do not consider the data on the ultraviolet fur coloration, because this is beyond my competences.

Experimental design

no comment

Validity of the findings

The part of manuscript related to the acoustic analysis of the calls of gliding or not gliding species of mammals has three main problems, which do not allow publishing the MS in its current form.
The first problem is the arbitrary and strange selection of a set of non-gliding species for inclusion them in analysis, and following this voluntary selection of papers on particular species. The gliding species are only among marsupials, rodents and colugos. Remains unclear, why the authors include in analysis primates and why the colugo (order Dermoptera) is included in Primates (Table S1). Unclear, why the authors do not include many species of rodents for which relevant data are available. Unclear, why the authors include in analysis reviews of old studies and ignore more recent studies, which provide substantially different values of acoustic parameters of the calls. Examples are provided below.
The second problem. During the formation the dataset for analysis (Table S1) the authors violate their own rule, of taking the absolute minimum and maximum of the fundamental harmonic of vocalizations. I checked two first papers cited in Table S1 (Martin, 2019; Bool et al., 2021): in one case, the authors took the values of dominant frequency; in the second case the authors took the values of the bandwidth frequency, although these values differ from call fundamental frequency in a few times.
The third problem. Both audio recording and the following acoustic analysis of the new vocalizations of the five species made unprofessionally. In one case, the authors used for recording an acoustic system which does not record the low frequencies; in the second case, they used the acoustic system which does not record the ultrasonic frequencies. Spectrograms of the "vocalizations" are created at very low resolution, which does not allow to see the acoustic structure of the vocalizations and is reminiscent of spectrograms of alien noises rather than mammalian calls. As a consequence, the values of the acoustic parameters of these "vocalizations" are incorrect. This is strange, because the second coauthor of this MS is a professional bioacoustics and, judging by previously published papers, can conduct correct bioacoustical analyses of mammalian calls. Call spectrograms should be created in acceptable quality and all analyses of calls from the five species of animals should be re-made for obtaining the real values of fundamental frequency.
These three main problems do not allow to consider the used for creating the statistical models data on vocalizations (Table S1) reliable and objective. As a consequence, there is no base to trust the results of the provided statistical models, because data manipulating (selective set of papers and inclusion in analyses of values of different frequency parameters) does not allow to obtain correct conclusions.

Additional comments

L. 21 We developed two datasets containing the vocal range (minimum-maximum; kHz).
Please make clear, for what frequency you provide the range of frequencies. Fundamental, dominant=peak or frequency of bandwidth? All these frequencies have different values in the same call.

L. 25-26 We found that gliding mammals used significantly higher frequencies than their non-gliding relatives.
Again, please make clear what frequencies do you mean.

L. 89-93 Aside from nocturnality and gliding, high-frequency calls are also commonly associated with various other traits: smaller body sizes (Martin, Tucker & Rogers, 2017; Newar & Bowman, 2020), open and structurally uncomplicated habitats (Boncoraglio & Saino, 2007; Ey & Fischer, 2009; Fischer, Wadewitz & Hammerschmidt, 2017), and higher degrees of sociality (Pollard and Blumstein 2012; Faure et al. 2017).
Please be more precise what do you mean under high-frequency calls. Which call types were compared to make this conclusion? Only ultrasonic or sonic and ultrasonic?

L 92 Wadewitz & Hammerschmidt, 2017
Is not in References

L 92-93 and higher degrees of sociality (Pollard and Blumstein 2012; Faure et al. 2017).
Irrelevant conclusion from irrelevant citing. Please delete.
The paper by Pollard and Blumstein 2012 is about the relationship between complexity of social structures and complexity of the acoustic communication.
The paper by Faure et al. 2017 is a comparison of ultrasonic communication in four strains of domestic mice.

L 99-100 Indeed, the reporting of USVs across several gliding mammal species suggests that USV production can evolve despite environmental constraints.
Please add reference

L. 167-169 Given their unique ecological niches and evolutionary pressures, we hypothesized that gliding mammals are more likely to exhibit crypsis in these traits than their close relatives.
Please indicate which taxa and which their close relatives you include to compare the acoustics and presence of ultraviolet. This remains unclear for the reader after reading the Introduction.

L. 178 The aim
The aim of this study should be provided at the end of Introduction. Now it is missing.

L. 183 summarized in Table S1
Please provide the titles for supplementary tables and legends for supplementary figures in files of these tables and figures. Now they are only present on web-site of PeerJ, inconvenient for the readers.

L. 185 multiple calls
Replace with multiple call types

L. 191-192 Across all published calls, we took the absolute minimum and maximum frequencies (kHz) for the final analyses (excluding harmonics; Newar and Bowman 2020).
Please indicate here that you took the absolute minimum and maximum fundamental frequencies (or the fundamental harmonic, as indicated in the legend to Table S1).

L 194 Arriaga & Jarvis, 2013
This reference is irrelevant for comparison of calls in pups and adults in mammals. Please replace with a relevant reference.

L 186-194 The databases used to search for these publications ... (Table S1)
The main question remains unclear: how did you select the species of non-gliders for inclusion in analyses? For which taxa? Why did you include primates, as there are no gliders within this order? If there are little publications on marsupials, there are a lot of on rodents. How did you select these studies? Why Table S1 uses many reviews and not on the original papers (where data are much more precise)?
For example, in Table S1 for Peromyscus californicus the Min-Max frequencies of 11.5-21 kHz are provided with reference to the review by García-Navas and Blumstein 2016 (The effect of body size and habitat on the evolution of alarm vocalizations in rodents. Biol J Linnean Soc, 118:745–751). In the review by García-Navas and Blumstein (2016) the mean maximum and minimum frequencies are given (remains unclear whether they are fundamental, dominant or bandwidth frequencies) for the alarm vocalizations of Peromyscus from paper by Miller and Engstrom 2012 (Vocal stereotypy in the rodent genera Peromyscus and Onychomys (Neotominae): taxonomic signature and call design. Bioacoustics, 21:193–213). In the paper by Miller and Engstrom (2012), many call types of Peromyscus are described, but among them there are no alarm calls. So, data used for Peromyscus californicus in Table S1 are non-valid. Furthermore, in a recent study by Riede et al. 2022 (Mechanisms of sound production in deer mice (Peromyscus spp.). J Exp Biol, 225:jeb243695) the high frequency for sweeps of Peromyscus californicus is 70-80 kHz.
Please provide the list of references for papers used in Table S1, because many of them are missing in the list of references to MS.

L 186-194 The databases used to search for these publications ... (Table S1)
Legend to Table S1: "the frequency ranges (minimum (kHz; the minimum frequency of the fundamental harmonic) – maximum (kHz; the maximum frequency of the fundamental harmonic)"
I checked two first in order publications and they did not fitted to the declared criteria for selection of the acoustic parameters.
For Acrobates pygmaeus, in Table S1 with reference to Martin 2019 are indicated Min-Max frequencies of 0.9-39 kHz. However, in the study by Martin 2019 there are no measurements of fundamental frequency, there are only measurements of the dominant frequency of the calls.
For Ailurops ursinus, in Table S1 with reference to Bool et al. 2021, the Min-Max frequencies of 0.19-13.5 kHz are indicated. However, in the study by Bool et al. 2021, there are no measurements of the fundamental frequency of the calls (chatters and clicks) but there are measurements of the dominant frequency and of bandwidth frequency (frequency range) of the calls, because there is no fundamental frequency in the chatters and in the clicks. The Table S1 provides Min-Max of bandwidth frequency, which contradicts to the author’s statement that the Table S1 contains measurements of the minimum and maximum of the fundamental harmonic.
Min-Max values of the frequencies of the frequency range (bandwidth frequency) can differ in a few times from the Min-Max values of call fundamental frequency. This makes doubtful the legitimacy of using the values provided in Table S1 for subsequent statistical analysis.

L 255 If a range was provided for the body mass, we took the average of the given maxima
This phrase is unclear, please re-write.

L. 314-315. low frequencies are ancestral for most mammals.
This is non-convincing result. At least in rodents, no adaptation of audition system is necessary to communication with ultrasonic calls, so probably this is a default state. At the same time, low-frequency hearing sensitivity needs in special morphological adaptations. See e.g., Mason, 2016. Structure and function of the mammalian middle ear. I: Large middle ears in small desert mammals. J. Anat., 228:284-299.

L 331-341 All five species were recorded with microphones sensitive to the human auditory range (20Hz-20kHz); however, we also opportunistically recorded sugar gliders (Petaurus breviceps) with ultrasonic detectors. We found that sugar gliders produce at least one ultrasonic call type along with three calls that extend into the ultrasonic range (broadband burst, sniffing, whistle) and three additional high-frequency calls (>10kHz) that could be detected on the ultrasonic microphones (bark, breathing, high frequency; Fig. S1-2 & Table S3). Of the four additional species recorded, springhares (Pedetes capensis) had the smallest and most consistent repertoire, with only one confirmed call type (growl) recorded over 250 times (Fig. S3 & Table S4). Yellow-bellied gliders (Petaurus australis) produced the longest vocalizations, with almost all calls lasting longer than one second, and had the most extensive repertoire (six call types; Fig. S4 & Table S5)
Acoustic analysis of calls of these five species has been conducted incorrectly.
Song Meter SM4BAT FS ultrasonic recorder with SMM-U1 ultrasonic microphone filters very strongly the frequencies below 8 kHz and therefore is inappropriate for recording and analysis of calls in the audible range of frequencies. This is visible even on Fig. S1. By this reason, frequency parameters of Bark, Broadband Burst, Breathing, Sniffing do not fit to reality.
The authors did not remove the pups (or a pair with pups) during automated recording of Petaurus breviceps. So, no base to statement that ultrasonic calls belong to adult individuals, not to pups of this species. Or these are not calls at all (I do not see the ultrasonic calls on spectrogram on Fig. S1). Moreover, the authors only found 2 calls (Table S3).
Audio Technica AT4022 omnidirectional microphones record calls only up to 20 kHz. So, if the species which Mr Anderson recorded have the calls in the ultrasonic range of frequencies, they could not anyway be recorded. When this material was collected? In which season of year? How many hours of recordings were done? How many recording sessions? What was the distance to animals? In which situations the calls were recorded? How did you select the calls for analysis from the entire massive of recordings? Where is bibliography for the paper Anderson 2022?
Spectrograms (Fig. S1, S3, S4 and S5) resemble most of all the spectrograms of background noise (especially Fig. S3) and noise from movements of animals and people. Spectrograms should be re-created to a format, allowing to see the acoustic structure of animal calls. The authors have the software Avisoft SAS Lab Pro, allowing to do this. The second coauthor of this MS is a professional bioacoustician and can do it.
The fundamental frequency of the calls should be measured manually (ask the second coauthor of this MS to do this), because the automatic measuring, conducted in the MS evidently provides impossible results (for example, 18-21 kHz for the Broadband Burst, although in reality the fundamental frequency of this call is less than 1 kHz).
FFT = 246. This is impossible. Please correct.

L. 341-343 Consistent with shortened vocal tracts in many marsupials (Charlton, 2021), yellow-bellied gliders, squirrel gliders (Petaurus norfolcensis), and common ringtail possums (Pseudocheirus peregrinus) produce low-frequency vocalizations
Please delete. Source-filter theory declares that sound source and sound filter are independent in mammals. Fundamental frequency of vocalizations depends on sound source. Formant frequencies determine the length of the vocal tract, the shorter is vocal tract, the higher are formant frequencies. These are basics of bioacoustics (see e.g., Taylor and Reby, 2010. The contribution of source–filter theory to mammal vocal communication research. J Zool, 280:221–236).

L. 347-351 Across all species in the dataset, we found that body size had a negative effect on the minimum (Estimate: x (±SE) = -0.52 ± 0.16; Effect Size: Fdf = 20.891,54, P < 0.001) and maximum (-0.43 (±0.09); Fdf = 18.551,54, P < 0.001) frequencies.
Validity of this result depends on proportion of how many ultrasonic and how many sonic (audible) call types were included in model. The audible calls in small mammals produced with vibration of the vocal folds commonly show the negative relationship of fundamental frequency with body size. The ultrasonic calls in small mammals mainly produced by the aerodynamic whistle mechanism can show positive, negative or no relationship with body size. The authors should indicate whether they mixed audible and ultrasonic call types in their model and if yes, to discuss this as limitation of the model in Discussion.

L. 366 Gliding mammals are a unique phenotype, with less than 50 extant species (Jackson, 2012).
This sentence belongs to Introduction.

L. 368-370
High-frequency acoustic signals are widespread among this group of species, which may be an indication that gliding mammals are more likely than non-gliding mammals to emit signals that are outside of the perceptual range of predators.
This statement too general to be reasonable. Most mammalian predators hear in high-frequency range and are dangerous for both gliders and non-gliders. Why calling at higher frequencies might be more important for gliders than for non-gliders?

L.457-461 Indeed, many species produce high frequencies and a variety of USVs in closed habitats despite these calls being easily absorbed by the spatially complex habitat in which they are produced. Therefore, while researchers have predicted that open habitats may be more conducive to the evolution of USV production, it may be that open habitats are better for recording USVs (as previously proposed by Newar and Bowman; 2020).
One of your study taxa, the rodents, produce USVs primarily in closed habitats and also in burrows underground (see e.g., Volodin et al. 2022. Ultrasound from underground: cryptic communication in subterranean wild-living and captive northern mole voles (Ellobius talpinus). Bioacoustics, 31:414-434).

L. 511-512
We found that gliding mammals emitted significantly higher vocal frequencies than their non-gliding relatives.
Most non-gliding species of rodents produce very high-frequency ultrasonic calls, some of them over 100 kHz. Most probably, the authors just did not include these species in their model.

Cite this review as

·

Basic reporting

Yes, the article is easy to read.
Yes, an overview of both ultrasonic calls and photoluminescence relating to gliders. I have suggested some additional references which may help in clarification.
Yes, this is very good. The structure makes it easy to find the different sections.
Yes, this is fine.
Yes, very good.

Experimental design

It is certainly novel – I don’t think anyone else was thinking of combining prevalence of photoluminescence with ultrasonic calls. I didn’t even realise a knowledge gap was there until they put the two together.
This seems very good. As the permit covers trapping and handling animals in the dark, it would be assumed that flashlights would be used on them as part of this process.
Yes, this is fine. I have requested a bit more detail in the wavelength of the flashlight used and the preservation status of the museum specimens.

Validity of the findings

Yes, the supplementary files contain this.
This is fine.

Additional comments

The dataset is well thought out, appropriate methods applied and the conclusions drawn are reasonable. From my point of view, perhaps the most valuable contribution of this manuscript is the comparison of photoluminescence in the live and dry-preserved specimens in the same conditions. The only important addition I would like to see made to this study is to include information on what chemical treatments (taxidermy, preservation, fumigation) the museum specimens might have gone through. Based on your comparison with a live individual, and observations in other studies, we know that studies of photoluminescence based on museum specimens can’t be taken entirely at face value. If you can give some kind of context to the pelt preservation, this would really help. If not, at least state that it is unknown what preservation processes were applied to the pelts. I have made minor comments below in order they appear in the text, especially in clarification of some of the background material.

Line 95–98: The background on forested habitats restraining USVs seems to be missing a mention of bats, which also echolocate in these environments. Bats that fly above the tree canopy tend to have lower-frequency calls than bats that fly down in the forest (see: Schnitzer et al. 2003 doi:10.1016/S0169-5347(03)00185-X ; or e.g. Hermans et al. 2023 https://doi.org/10.1186/s40462-023-00387-0 )
Line 112: You might want to add in one of the tryptophan metabolite references to the list, either one of Rebell’s, or Nicholls and Rienits 1971 https://doi.org/10.1016/0020-711X(71)90031-0
Line 113–116: Yes, porphyrins are extremely photodegradable, but tryptophan metabolites can take months to degrade in the sun (Schäfer et al. 1997 https://doi.org/10.1111/j.1478-4408.1997.tb01862.x ; or Posudin 2007 https://doi.org/10.1201/9781482294507 ; or Longo et al. 2013 https://doi.org/10.1111/ics.12054 ). [Some photoluminescent animals such as sheep, horses and gerbils are diurnal (on line 237–238 you touch on this). It’s only the bright pink-orange-red photoluminescence, probably from porphyrins, that we wouldn’t expect to find in diurnal mammals.] Leave the Reinhold et al. 2023 citation out of that sentence, because that’s not at all a prediction that was made for photoluminescence in general.
Line 119: Leave the citation out of that sentence, or use one of the direct citations of melanin masking instead – Rebell 1966 would be a good one ( https://www.nature.com/articles/209913a0 ). I would also leave the word ‘nocturnal’ out of that sentence, as it makes it confusing.
Lines 125–128: Marshall and Johnsen 2017 had five criteria, the fifth being that there are visual behaviours that would be assisted – add this in if you think it’s appropriate.
Lines 130–131: The excitation range for visible-spectrum photoluminescence emission is broader than that. Porphyrins in particular span the upper end of the ultraviolet and into the violet, as well as lesser excitations up into other colours (Hamchand et al. 2021). Some studies show peaks for protoporphyrin just below 400 nm, but Huang et al. 2010 (doi: 10.1016/j.talanta.2010.07.034 ) shows the peak absorption of three porphyrins (that are all found in fur) at just above 400 nm. So the restriction to 400 nm is arbitrary. Also add the words ‘emission’ and ‘excitation’ to that sentence to clarify.
Lines 132–139: This section about UV light needs refocusing. Remember, it’s not the excitation wavelength that animals need to see, but the emission wavelength. Photoluminescence turns ‘invisible’ light into ‘visible’ light. So animals don’t need to see the original light source, only the photons after they are re-emitted at longer wavelengths. Maybe just cutting lines 132 to 135 out could save confusion. The point should be that there is ample excitation light in sunlight (Marshall and Johnsen 2017), moonlight has been tested experimentally (Kloock 2005), then there’s twilight… [So maybe the filtering in the eyes of diurnal mammals cuts out the glare and enables them to see the photoluminescence in the same way you use a yellow filter on your camera (see mention of yellow lenses in Marshall and Johnsen 2017), or the clear lenses of nocturnal mammals let more light in, meaning they might pick up a faint glow, if you want to include a discussion about filters.]
Lines 174–176: This is just a suggestion to be more specific, but you can choose to leave it out and keep it as a more general prediction: “However, given the photodegradability of some porphyrin luminophores that accumulate in fur and the communication potential of UVP in nocturnal species, we also predicted that pink-orange-red UVP would be strongly associated with nocturnality and sociality.” Or keep it general, but add that you expected it to be particularly so for pink-orange-red. [Have a think about why there would be a greater communication potential of UVP in nocturnal mammal species – sunlight gives more opportunity for excitation than starlight or moonlight, and other studies suggest that diurnal birds may use it – why nocturnal for mammals? Or you could just say it was hypothesised by Kohler et al. 2019 and Olson et al. 2021.]
Lines 202–203: There is some vocalization data published for Petaurus australis (e.g. Whisson et al. 2021 https://doi.org/10.1371/journal.pone.0252092 ), though not ultrasonic.
Lines 210–211: I would presume the museum specimens were all dry-preserved study pelts. But, it’s really important to spell out what sort of specimens they were, if any were preserved in ethanol. Also, try to find out if you can the taxidermy processes the specimens have been through. If they have just been salted and dried to be kept for scientific study, that’s fine. But if the pelts have been washed or soaked in any tanning chemicals, or undergone any treatments for insect infestation, that’s important to know too, as it could chemically alter the luminophores in the fur. If any of your specimens were exported from one country then imported into Canada, it may be important to know what fumigants would have been used on them (e.g. carbon disulfide or methyl bromide because of the possibility of imparting bright cyan or green photoluminescence, separate from the interesting problem you had with the appearance of cyan as an artefact of photography). Adding this data may give an indication as to the level of false negatives and false positives of photoluminescence.
Line 218: You need to say what the wavelength (or wavelength range) of the flashlight was. This will give an indication of the types of luminophores it would be expected to activate. I think it is 395 nm, but you need to state this.
Line 232: Striped possum, not striped opossum.
Line 231 – 233: Latham 1953 only described a “vivid lavender color”, not white.
Line 234: Cut out the mention of ‘green’ – this was a colouration in birds, so not necessarily in fur.
Line 237–239: I think what you’re saying here is that human hair doesn’t have that emission peak around 330 nm that the other fibres do. Yes that would be an ultraviolet emission, but you’re only talking about visible-spectrum emissions over 400 nm (animals would need UV vision to see the 330 nm emission peak). Sorry for the confusion. So better to combine those two sentences to say something like: Human white hair emits a blue color, as does the pelage of minks, rabbits, goats and sheep which Millington (2020) determined to be photoluminescence, due to either tryptophan metabolite luminophores or protein deep-blue autofluorescence.
[Collins 1992 ( https://openaccess.city.ac.uk/id/eprint/7894/1/ ) goes into more detail about colourless phosphorescence etc. of sheep’s wool on pages 56, 59 and 60, attributing it to actual phosphorescence (by an unknown phosphor excited at 340 nm), not some other effect of light. No need to cite this, just for your information. Also, when the glow is dim, some colours are lost, so we only see it as whitish. So definitely keep in your Supplementary Table including white photoluminescence.]
Lines 240–241: Just to clarify – were you trying to exclude white photoluminescence that might have excitation wavelengths >400 nm, or was it to dismiss other effects of white fur catching the light?
Lines 309–312: If you want photoluminescence data for greater gliders, they were only photoluminescing pale grey and white (in a dark lab) on their light-coloured parts of fur – you can go to page 146 of Reinhold 2023: https://doi.org/10.13140/RG.2.2.12146.48325 (other gliders - feathertail and Krefft’s - are listed on pages 142 and 144)). I listed them as Petauroides volans, but under the split they would be Petauroides minor. Apparently they don’t really call, at least not loudly, which is interesting in itself.

Lines 361–362: You could add here that this diurnal species exhibited both pink and blue (but then, was the pink in fur that would have been protected from the sun?).
Line 467: Krefft’s has two ‘f’s.
Lines 466–467: …and Hylopetes spadiceus (red-cheeked flying squirrel) and Pteromyscus pulverulentus (smoky flying squirrel) (Toussaint et al. 2023).
Lines 471–473: What colour of photoluminescence – pink or blue?
Lines 473–475: Well done on testing this in the same conditions! I am so glad someone has finally done this. This gives a great indication of the likelihood of false negatives in museum specimens. Was the ventral museum specimen pictured in Figure 2 the particularly weak one? Given the variation in museum specimens, it would be good to give a sample size for each species you tested, either in the methods or in Table S1.
Line 477: You could add a sentence here something like: Due to the photosensitive nature of porphyrins, reddish photoluminescence is generally not expected to be retained in museum specimens (Hill 2010).
Line 483: Take the Reinhold et al. 2023 citation out of this list – only porphyrins would be associated with nocturnality, not tryptophan metabolites, given that the latter kind of photoluminescence is known to occur in several diurnal mammals.
Lines 487–489: Again, see discussion on lines 113–116. There is no reason that non-porphyrin photoluminescence wouldn’t be common in diurnal mammals, which makes both diurnal and nocturnal mammals likely to display it. Photoluminescence has certainly been recorded in several diurnal species.
Supplemental Table S1: A great table, with all the raw data easily accessed. A couple of additions: Photoluminescence in the fur of Trichosurus vulpecula has been published: Bolliger 1944 and Nicholls and Rienits 1971. You could also add in Toussaint et al.’s 2023 records of pink photoluminescence in two species of south-east Asian flying squirrels.

Reviewer 3 ·

Basic reporting

The paper needs to be tightened up a fair bit so it is clear what is going on.
• Throughout the manuscript there a comma after the authors name and before the year in many places but not in others. E.g. Reinhold et al., 2023 and Reinhold et al. 2023 are referred to.
• It needs to be clear which species are gliders and which are non-gliders throughout.
• When referring to the non-gliders it needs to be clear why they are included. E.g. members within the same family as gliders or the most closely related family.
• Need to cross check all references in the text with the reference list. There are many changes needed
• There are references in the results section – these should be removed and only results included
There are many specific edits in an attached file.

Experimental design

The aims and scope are ok

Validity of the findings

Findings are ok

Additional comments

See attached file

Annotated reviews are not available for download in order to protect the identity of reviewers who chose to remain anonymous.
Cite this review as

---

## Round 0.2 · Minor Revisions

The reviewers continue to be wide apart. I hope you will carefully inspect their comments. It appears reviewer #2 believes the paper is publishable with one more round of minor revisions. Since Reviewer # 1 is quite dissatisfied still, I have taken the unusual step of initiating a search for a third reviewer willing to comment. However reviewers may be hard to find at this season to I suggest you go ahead and submit revisions addressing reviewer #2's comments. If I receive a version meeting these before January 1 or the potential third reviewer's comments, I will make a final decision then.

Reviewer 1 ·

Basic reporting

Idea of this study is very interesting, but the study itself was again conducted with cruel mistakes. The authors revised the manuscript and responded to many questions. The problem of selection of a set of non-gliding species for inclusion them in analysis is now cleared and does not evoke any questions. However, two other main problems of the manuscript were unresolved: extraction of the values of “frequency” parameters prom published papers for preparing the dataset for the statistical analyses (the first problem) and conducting adequate measurements of vocalizations of five mammalian species (the second problem). This study pretends on meta-analysis, but the provided comparison is unconvincing and misleading. If published without substantial reworking, it will confuse the readers unfamiliar with bioacoustics. If the two mentioned problems will not be resolved I recommend to remove the acoustic part from this study and to publish alone the ultraviolet part, for which the referee experts are more positive.
Structure of the manuscript is very complicated, because, in addition to the main file, there are many supplementary files, what is difficult for reviewing. Supplementary tables now have the headings (bravo), but supplementary figures still have not legends and to understand which call originates from which individual animal, the reviewer should search the corresponding places in the text. I am unfamiliar with detailed rules of the journal, but for reviewers (and for further readers too) will be much more convenient, if all the supplementary files (for the exclusion of permissions) will be collected in one single integrated PDF file. The authors can also make such integrated file additionally to the dozen of provided supplementary files.

Experimental design

Manuscript represents a mix of data extracted from literature and data from own measurements of the authors. All these data together are used for creating statistical models. Such mix of data imposes limitations on quality of presenting own measurements, because they are branching to the main line of the manuscript.
In spite of the extensive supplementary materials, many questions remain unclear: conditions of recording of the 5 species of animals (in captivity or in the wild, on which devices, by three different researchers; situations in which these calls were recorded; data volume (how many hours of recording, number of animals, number of recorded calls; selection of calls for analysis from the total massive of data; conducting the spectrographic analysis (there are no illustrations of measured acoustic parameters and even their description). Separation of calls to different types is not substantiated. And, of course, the obtained result (Tables S3 and S4) cannot pretend on "description of species vocal repertoires", as it was claimed in the manuscript. I think that in the massive of treated papers the authors will easily find examples of good descriptions of vocal repertoires for different species of animals.
For the manuscript, it would be better if this branching line with measurements of collected by the authors non-systematically calls of the five species was deleted. This would make the manuscript more solid and clear. I understand that because of insufficient data on calls of the five “new” species they cannot be published separately (they can only be hidden in Supplementary materials which are rarely looked by the readers). But it is necessary at least to conduct an adequate spectrographic analysis of these calls, to substantiate their separation to types and make normal visualization (see below).

Validity of the findings

In spite of the author claims (L 21-22, L 213-214) that in the revised version all extracted parameters from Table S1 were checked and now the Table only contains the values of call dominant frequency (the lowest and the highest across different call types), in reality this claim is wrong.
As the authors evidently have difficulties with understanding what different "frequency" parameters mean, I prepared and attached a PDF file with the guide on "frequency" parameters in the spectrographic software (file Frequency.pdf). Please use it for the real checking of the extracted values. As one line of Discussion is related to call "high-frequensiness", so, the optimal parameters for estimating this thing are the values of fundamental frequency. This is a single parameter of "frequency", which responds for whether animal call will be "high-frequency" or "low-frequency". All other acoustic parameters are not related with call "high-frequensiness".
As in the previous version of manuscript, the values of acoustic parameters of "frequency", introduced in the statistical models (Table S1) does not deserve any trust. From the papers, they were selected absolutely voluntarily. For example, in spite of presence of measured values of dominant frequency in the papers, the authors introduced in Table S1 the values of fundamental frequency or the frequencies of bandwidth. Some measurements are not even clear where they were taken from (were they invented by the authors?). One gets the impression that the authors either do not understand anything about the acoustic parameters they are writing about or deliberately deceive the reader by arbitrarily choosing those values from the published data that suit them best.
Randomly chosen examples of the extracted values of "frequency" from the Table S1 (the authors claim that these are the values of call dominant frequency) are given below:

Acrobates pygmaeus, Martin 2019, Min-Max frequencies of 0.9-39 kHz
In Table S1 the authors provide the values of fpeak (= dominant frequency). But these values are not the mean values for different call types (with the lowest and the highest values of fpeak), but the lower and upper limits of these values, measured in both cases in the single call. These values can be outlier values. Min-max mean values of fpeak across different call types are 3.4-29.2 kHz.

Ailurops ursinus, Bool et al. 2021, Min-Max frequencies of 0.19-13.5 kHz
In Table S1, the Min-Max of bandwidth frequency is provided (see the attached file). In the paper by Bool et al. 2021 there are measurements of fpeak. Min-max mean values of fpeak across different call types are 2.77-6.32 kHz.

Dasyrus hallucatus, Dempster 1994, Min-Max frequencies of 0.4-2 kHz
The Table S1 provides the frequencies which were unclearly how selected. The paper by Dempster 1994 provides only the lower and upper frequency limits of entire calls (see the attached file). Measurements of fpeak are missing. The values of 0.3-2.1 kHz from Table S1 fits to frequency limits of call type hiss. However, the paper describes other call types, for instance, sniff with values of frequency limits of 0.7-9.5 kHz. Why the authors selected for Table S1 the values of frequency limits (but not declared fpeak) and why of particularly of the hisses remains unexplained.

Petaurus australis, Whisson et al. 2021, Min-Max frequencies of 0.5-4.5 kHz
The Table S1 provides the values of f0min and f0max, estimated from the spectrogram figure (what was not indicated by *). In the paper by Whisson et al. 2021 there are no measurements of the calls. However, on the Fig. 1A and 1B is visible that fpeak of different call types is lying within 2.5 kHz.

Cynomys gunnisoni, Ackers and Slobodchikoff 1999, Min-Max frequencies of 0.70-6.23 kHz.
These values just invented by the authors. The paper by Ackers and Slobodchikoff 1999 did not measure any calls. On the figure in the paper there is a spectrogram with "dominant harmonic frequency" ranging approximately from 2.4 to 3.5 kHz.

Cynomys gunnisoni, Loughry et al. 2019, Min-Max frequencies of 0.38-1.19 kHz
The Table S1 provides the values of f0min and f0max for the alarm barks, which are measured in the paper Loughry et al. 2019. There are no measurements of fpeak in the paper by Loughry et al. 2019.

Cynomys ludovicianus, Waring 1970, Min-Max frequencies of 0.1-4 kHz
The Table S1 provides the min value of frequency range (0.1 kHz), 4 kHz was invented by the authors. In the paper by Waring 1970 the measurements of f0min and f0max and min-max frequency range are given (in case of the missing f0, as a rule 0.1-8.0 kHz) for many call types. There are no measurements for the fpeak. For the lowest-frequency call type screem, the f0min = 0.3 kHz.

Glaucomys sabrinus, Gilley et al. 2019, Min-Max frequencies of 10.89-24.89 kHz
The Table S1 provides the values for the lowest and upper frequencies of bandwidth. In the paper by Gilley et al. 2019, the measurements of fpeak (termed Fmax) are provided, which vary from 18.09 to 20.67 kHz between call types, bur the authors did not use them.

Glaucomys volans, Gilley et al. 2019, Min-Max frequencies of 15.55-25.62 kHz
The Table S1 provides the values for the lowest and upper frequencies of bandwidth. In the paper by Gilley et al. 2019 the measurements of fpeak (termed Fmax) are provided, which vary from 18.09 to 18.30 kHz between call types, but the authors did not use them.

Glaucomys sabrinus, Murrant et al. 2013, Min-Max frequencies of 20-80 kHz
Both values are just invented by the authors. In the paper by Murrant et al. 2013, for G. sabrinus Type 1 call type was described with fpeak = 60 kHz (not 80 kHz). Calls of Type 2 with f0min = 18.5-20 kHz are described only for G. volans.

Glaucomys volans, Murrant et al. 2013, Min-Max frequencies of 19-80 kHz
The Table S1 provides the values for f0min of call type 2 (18.5-20 kHz). The value of Max frequency of 80 kHz was invented by the authors. The paper by Murrant et al. 2013 describes the calls of Type 1 with fpeak = 60 kHz (not 80 kHz) for G. volans.

Sicista subtilis, Volodin et al. 2019, Min-Max frequencies of 6.21-9.86 kHz
The Table S1 provides f0min and f0max, with minimum and maximum values of these parameters across 11 individuals. The paper by Volodin et al. 2019 contains the measurements of fpeak. The mean values of fpeak vary from 7.90 kHz for females to 8.14 kHz for males.

The second problem is the conducting the adequate spectrographic analysis of recorded calls for the five species of animals. I keep a pair of Petaurus brevicers at home as pets. And there were no problems to make a few recording of their calls (see the attached file). However, these calls are perfectly incomparable with calls described in the manuscript (Table S3 and S4, Fig S1, Vocalizations in Results). The Table S3 contains impossible values of call "frequency" (in a few times more than real). Fig. S1 shows blurry coloured blots, not call spectrograms. Compare them with spectrograms of calls from Petaurus brevicers in the attached file, on which you can see all nuances of the acoustic structure of the calls. If you cannot to create and to display to the readers similar figures, just delete your measurements, because you created and measured spectrograms of noise.
If the authors will nevertheless decide to keep their data on measurements of recorded calls of the five animal species, all their spectrographic analysis should be re-made. The authors should re-make all spectrogram figures so that the acoustic structure of the calls was visible. This will allow to understand what was indeed measured in these calls. The authors should clearly indicate, which parameters and how were measured. The authors write (Table S3 and S4), that "Frequency estimates taken from the dominant harmonic, if harmonics present". However, the harmonics are only visible on Figs 2SB and 2SE. And remains unclear, which of these harmonic is dominant. As the Fig. S2B displays the yap of Petaurus brevicers, so the provided values (0.75 – 2.53 kHz) do not fit to the real peak frequency. If you measure the peak=dominant frequency, so please show on the figure the power spectrum and position of the peak frequency, it is absolutely non-evident and non-measurable from spectrogram (see Fig. 3 in the attached file). If the authors are uncapable to conduct the accurate and adequate spectrographical analysis of the calls (although Irena Schneiderová previously made rather good papers on bioacoustics), I recommend to delete this part from the manuscript.

Additional comments

Without resolving the main problems of re-making the table with the acoustic parameters for the statistical analysis and conducting professionally competent acoustic analysis of the new vocalizations of the five species I do not see any sense to edit the text of the manuscript. This had to be done at the next round of revision.
A few particular comments (Lines are provided by the clean variant):

L 21-22 minimum-maximum of the dominant frequency; kHz
L 26 maximum dominant frequencies
L 213-214 absolute minimum and maximum frequencies (kHz) of the dominant frequency
In the dominant=peak call frequency there is only one meaning. There are no the minimum and maximum dominant frequencies in a call.

L 105 with no effect on the maximum frequency
There no just "frequency" in the calls. Which frequency?

L 213-215 Across all published calls, we took the absolute minimum and maximum frequencies (kHz) of the dominant frequency for the final analyses (this often corresponded to the fundamental harmonic, if harmonics were present; following Newar & Bowman, 2020)
No, the Table S1 still contain any values of call "frequencies" deliberately or randomly taken from the published data (see above).

L 213-215 Across all published calls, we took the absolute minimum and maximum frequencies (kHz) of the dominant frequency for the final analyses (this often corresponded to the fundamental harmonic, if harmonics were present; following Newar & Bowman, 2020)
In short-snouted mammals, the peak frequency of the calls, as a rule, exceeds a few times the fundamental frequency. This is related to position of the first formant frequency (which depends on the length of the vocal tract, the shorter is tract the higher the formant frequencies. Even in men, the fundamental frequency of voice is about 150-180 Hz, but the dominant frequency of some vowels is 500 Hz, what fits to the area of enforcement of the first formant.

L 215 following Newar & Bowman, 2020)
This is preceding research of the same authors. If in the current research the authors show full non-understanding of sense of used by them acoustic terms, so in the preceding research extraction of data from the papers was done in a similar way. It’s pity.

L 215-219 For noisy calls, such as broadband calls, where the dominant amplitude is spread across a range of frequencies, we took the minimum and maximum frequencies of the amplitude peak which corresponds to a larger dominant frequency range than tonal calls (this was only done when amplitude information was provided; see Figure 1A in Newar and Bowman 2020).
What is dominant amplitude? There is no such term.
minimum and maximum frequencies of the amplitude peak. Judging by the figure, "amplitude peak" is call dominant frequency of the broadband call. The value of dominant frequency can easily be measured in any software. What is "the minimum and maximum frequencies of the amplitude peak"? Peak has only one measurement (this is peak!). How did you measure "the minimum and maximum frequencies of the amplitude peak"? Judging by Table S1, the authors just looked on the figure and wrote any digit?

L 369-372 All five species were recorded with microphones sensitive to the human auditory range (20Hz-20kHz); however, we also opportunistically recorded sugar gliders (Petaurus breviceps) with ultrasonic detectors
The used ultrasonic detector filters out the lower 5 kHz during recording. So, the values of all calls of Petaurus breviceps (aside the ultrasonic calls), measured on such recordings, are incorrect and exceed the real values in a few times.

L 385, L 387 Fig. S5
There is no such figure in the revised variant, only Fig. S1 and Fig. S2. Please make the legends to the supplementary figures, better in one file (PDF).

Supplemental Information
Captive Sugar Gliders Recordings We suspended an omnidirectional SMM-U1 ultrasonic microphone (Wildlife Acoustics) above the enclosures approximately 1-4m from the end of the microphone. We connected the microphone to a Song Meter SM4BAT FS ultrasonic recorder (gain = 12dB, sampling rate = 192kHz, 16-bit resolution; Wildlife Acoustics)
Please indicate frequency response for the recording system, from 5 kHz to 96 kHz. This explains very high values of dominant frequency in the calls of Petaurus breviceps.

Captive Springhares Both species were recorded with a Marantz PMD 662 recorder
Please indicate sampling rate

Free-Ranging Marsupials Recordings were made using two Audio Technica AT4022 omnidirectional microphones attached to a Fostex FR2-LE Field Recorder
Please indicate sampling rate

Fig. S2
On the figure, filtration of low-frequency noise is clearly visible. This can affect the values of dominant frequency. Please indicate parameters of the filtering (range, filter).

Table S1. Summary of references for the frequency ranges (minimum (kHz; the minimum frequency of the fundamental harmonic) – maximum (kHz; the maximum frequency of the fundamental harmonic))
Frequency of the fundamental harmonic is not the dominant frequency! These are different acoustic parameters. They are measured differently and as a rule have different values Please see the attached file.

Table S3 and S4 peak frequencies represent the frequency with the highest entropy.
The peak frequency represents the frequency with the highest energy (in the analysed fragment of spectrogram), not entropy!

Annotated reviews are not available for download in order to protect the identity of reviewers who chose to remain anonymous.
Cite this review as

·

Basic reporting

No further comments.

Experimental design

No further comments.

Validity of the findings

No further comments.

Additional comments

The authors have put in a pleasing effort to address the first review. Although I hesitate to give credence to results from museum specimens where the light exposure and chemical treatment histories are unknown, the authors have made an effort to find this out, and stated the use of museum specimens as a limitation. It is particularly refreshing that they have used photographs illustrating the difference in intensity between natural photoluminescence in live animals, and the severe fading of this in dry specimens. This is an important contribution in comprehending the scale of false negatives in museum studies. I can only comment on the photoluminescence part of the study. My apologies for not picking up some of these more detailed corrections in the previous draft.
My suggestions below are not in order of importance, but sequential through the manuscript.

Tracked changes manuscript
Introduction
Paragraph lines 143 to 167: Thanks for clarifying the excitation wavelengths – otherwise the term ‘ultraviolet-induced photoluminescence’ is overly restrictive, but since your flashlight was just under the ultraviolet/violet threshold, justified for this study in which you only tested at that wavelength. Your elaboration tells the reader that it’s not necessarily ultraviolet-induced in nature, but ultraviolet-induced in experimental conditions.
Just to be clear, it doesn’t matter whether animals can see the ultraviolet-violet-blue excitation wavelengths, only the emission wavelengths, which, for the kind of photoluminescence we’re talking about, occur in the visible part of the spectrum. When you say “re-emitted light”, meaning “photoluminescence”, clarify that this light is now in the visible spectrum, as it reads as if it’s just re-emitted UV. The excitation wavelengths are only important for the ambient light provided in the environment, and whether or not animals can see this excitation light is irrelevant to the viewing of emitted photoluminescence. If the point you are making is about what colour wavelengths terrestrial vertebrates can see, have another read of the paragraph under (c) Spectral sensitivity ranges of potential viewers in Marshall and Johnsen 2017. Humans are good at seeing fluorescence because of their sensitivity to green. In fact, most terrestrial vertebrates have peak sensitivity at green (but we, and presumably many others, are not as sensitive to reddish photoluminescent emissions). The photoluminescence we are talking about (pink, blue) doesn’t emit in the ultraviolet; it emits in the blue to pink parts of the spectrum, so ultraviolet sensitivity isn’t directly relevant. The ‘ultraviolet’ in photoluminescence isn’t like the ultrasound crypsis of vocal calls – the former is an excitation and the latter is an emission – and for the purposes of this paragraph it may be more useful to think of photoluminescence as visible-emitted rather than ultraviolet-induced. The photoluminescence we are talking about emits in colours in the visible spectrum, so animals only need to see in colour, or detect the contrast of the glow. At night, there would need to be strong enough illumination to set it off, and the eyes of the viewer sensitive enough to see it. In the day, the excitation light will be strong enough – it just depends on the ocular filtering of the viewer to cut out reflectance. Sorry to go on – this is all just context for you to re-read the paragraph from Lines 143 to 167 and see if you could make it make more sense.
Line 190: If you add “or hypotheses” after “evidence”, then that would give you a reason for testing UVP when there isn’t technically any evidence that it hides animals from predators in a terrestrial environment.
Materials & Methods
Line 222 reads as if you list 70 species of gliding mammal, but in Table S1 there are only 15. So I think the 70 were mammal species in general? If would be helpful to state in the text how many species were gliding and how many weren’t, or how many non-gliding counterparts you used for each gliding mammal species.
Results
Line 437: Double-check you mean that your final dataset contained 78 species, not 82 or 83? It would be helpful just to clarify again whether the photoluminescence in the 21 species was all photoluminescence including white, or just blue-green-yellow-pink.
Line 442: It would be useful to say after the sentence about the number of species exhibiting UVP, that out of the 25 photoluminescent nocturnal species, 22 displayed pink photoluminescence. (This is important because it is the porphyrins that are tied to nocturnality because of photobleaching, not the tryptophan metabolites.)
Discussion
Line 457: It would be helpful to again specify that the UVP you are talking about is for blue-green-yellow-pink, or non-white, just to remind readers that the white was not analysed, as this would not normally be expected.
Lines 458 – 460: You need to specify that the photodegradation of luminophores is for porphyrins, or pink-orange-red photoluminescence, or pink-red photoluminescence which is suspected to be from porphyrin luminophores.
Line 586 – 594: Consider elevating some of this discussion on the difference in photoluminescence between live and museum specimens to the Results. I know you were only looking at presence/absence, but the results of this comparison you did have such a bearing on the interpretation of museum specimens for both pink and blue photoluminescence and the potential for false negatives. Not essential, but I did feel it was missing from the Results (and there’s a helpfully illustrative figure for it), perhaps an editorial decision. In Figure 2, for (B), specify that they’re dry-preserved (if you haven’t done this already – I can’t see the updated figures in this version).
Line 602: should be “natural photoluminescence”, not “true photoluminescence”.
Line 610: I would take out “by-product” or make it that they may be a by-product, or that it’s just through various physiological pathways – we just don’t know yet whether they end up there as a by-product or if they serve a specific function (not necessarily optical) in the fur. And if they are a by-product, then perhaps they’re not selected for on their potential camouflage functions, which doesn’t support the hypotheses you’re looking at.
Line 614: Add the word “as” before photodegradable – they still photodegrade, but over several months, not minutes.
Lines 617 – 619: I’m not quite following this sentence – there hasn’t really been direct comparisons of tryptophan metabolite concentrations in diurnal vs nocturnal species with sufficient sample sizes of either – maybe you mean one or the other has darker fur, but then you can’t necessarily tell how concentrated the luminophores are when they are masked by melanin. You would expect more porphyrin-based photoluminescence in nocturnal species because of photobleaching, but no reason for it so be so for tryptophan-based photoluminescence. If you’re talking about photoluminescence in general, then yes, but only because of porphyrins.
Line 621: For the mention of the diurnal degu to make sense, it needs to say whether it had dark or light fur.
On reading Chávez et al. 2003, I see no mention of pink and blue photoluminescence in the degu, only a general UV reflectance. Please re-read this reference and decide if you really want to cite it in the context of your work. I would not count it as a case of fur photoluminescence.
Line 623: Earlier you updated these numbers to 25 (+ the white ones) nocturnal species that showed UVP, and 21 that didn’t (Line 442) – might need to double-check the counts.
Line 637: Add “potential” to “form of crypsis” – there’s no evidence that it is a form of crypsis. Here you say that UVP is not widespread across gliding mammals. It seems from Table S1 that out of the 15 (14) gliding species listed, 7 photoluminesced? Whatever the numbers are, it would be handy to state what numbers you are drawing this result from. You could break it down by colour (if pink has been hypothesized as Batesian mimicry of owls, and blueish has been hypothesized as camouflage against lichens). I think all of the photoluminescing gliding mammals showed pink except for the greater glider. This is a pretty interesting finding in itself. About half of all your nocturnal species photoluminesced, and so did about half of your gliding mammal species.
Conclusions
Considering the word “crypsis” is in the title, I would like to see a round-up of how this ties in in the Conclusion, even if the study didn’t turn out to be as much about crypsis as about other things.
Line 646: Again, consider specifying that the 10 species showing UVP showed coloured UVP, as the general reader would expect this to include white.
Line 647 – 648: I wouldn’t really call the finding that luminophores are responsible for photoluminescence a hypothesis – it has been demonstrated repeatedly from the 1950s onwards from laboratory extractions. (And most of the work was done on a diurnal species.) I would mention in the middle of this sentence that about half of your nocturnal species showed photoluminescence, to give some meaning to how likely it is that a nocturnal mammal will have UVP (as you had initially predicted that UVP would be found in most nocturnal mammals). Another round-up sentence about how gliding mammals showed more, the same, or less, UVP than other nocturnal species would give more closure.
Table S1
I think the captive sugar gliders over there originated from West Papua, so according to the recent species split, may now be classed as Krefft’s gliders. Check if your specimens of Petaurus breviceps are now called P. notatus, in which case the vocalization and UVP data of these two “species” would be combined in to the one species, P. notatus.
Pine et al. (1985) documented fluorescence in Marmosa mexicana [if you can’t get the whole paper on Google Scholar, get it from ResearchGate]. Check if there are also species listed in Pine that may have had a taxon change since, such as Metachirops opossum.
If you want a more detailed citation for Didelphis virginiana, use Tumlison and Tumlison 2021.
Mus musculus was documented as being non-fluorescent by both Rebell 1966 (insignificant amount of kynurenine in fur compared to rats) and Tumlison and Tumlison 2021.
Udall et al. (1964) also documented the photoluminescence of Rattus rattus as being green-blue.
There are some species where you have only listed the pinkish photoluminescence, not the other colours: Dasyurus hallucatus (yellow, green, Reinhold 2023), Trichosurus vulpecula (blue, Bolliger 1944; blue, Reinhold 2023), Perameles nasuta (blueish white, yellow, Reinhold 2023), Petaurus notatus (blueish white, Reinhold 2021; blueish white, Reinhold 2023). Even if the pink is the most noticeable, the distinction between pink and other colours is quite important because they are probably a different class of luminophore. Reinhold (2023) lists Dactylopsila trivirgata also as blue-white, not just white.
Petauroides volans is missing the data on nocturnality etc.

Sorry to be so picky – all in all you’ve done an excellent job!

---

## Round 0.3 · Minor Revisions

Most changes requested by reviewer #2 have been made. The editor requests that you fix the following in a v3:
Line 133: "perhapsmore" to "perhaps more"
This should take a few seconds!

---

## Round 0.4 · accepted · Accept

Thanks for the final change.